TOOLS

# Membrane contact site detection (MCS-DETECT) reveals dual control of rough mitochondria–ER contacts

Ben Cardoen[1]*, Kurt R. Vandevoorde[2]*, Guang Gao[2]*, Milene Ortiz-Silva[2]*, Parsa Alan[2], William Liu[2], Ellie Tiliakou[2], A. Wayne Vogl[2], Ghassan Hamarneh[1]**, and Ivan R. Nabi[2,3]**

**Identification and morphological analysis of mitochondria–ER contacts (MERCs) by fluorescent microscopy is limited by subpixel resolution interorganelle distances. Here, the membrane contact site (MCS) detection algorithm, MCS-DETECT, reconstructs subpixel resolution MERCs from 3D super-resolution image volumes. MCS-DETECT shows that elongated ribosome-studded riboMERCs, present in HT-1080 but not COS-7 cells, are morphologically distinct from smaller smooth contacts and larger contacts induced by mitochondria–ER linker expression in COS-7 cells. RiboMERC formation is associated with increased mitochondrial potential, reduced in Gp78 knockout HT-1080 cells and induced by Gp78 ubiquitin ligase activity in COS-7 and HeLa cells. Knockdown of riboMERC tether RRBP1 eliminates riboMERCs in both wild-type and Gp78 knockout HT-1080 cells. By MCS-DETECT, Gp78-dependent riboMERCs present complex tubular shapes that intercalate between and contact multiple mitochondria. MCS-DETECT of 3D whole-cell super-resolution image volumes, therefore, identifies novel dual control of tubular riboMERCs, whose formation is dependent on RRBP1 and size modulated by Gp78 E3 ubiquitin ligase activity.**

## Introduction

In the cell, organelles communicate with each other at membrane contact sites (MCS) where two membranes come in close proximity, as close as 10–30 nm, without fusing (Helle et al., 2013; Valm et al., 2017). Mitochondria–ER contacts (MERCs), an MCS subclass, are hubs for exchange between the ER and mitochondria, enabling calcium transfer required for mitochondrial enzyme activity and ATP production, phospholipid and sterol biosynthesis, mitochondrial dynamics and metabolism, as well as the execution of cell death programs (Rowland and Voeltz, 2012). MERCs are closely associated with disease progression, including cancer, neurodegenerative, cardiovascular, and other diseases (Barazzuol et al., 2021; Díaz et al., 2021; Markovinovic et al., 2022). MERCs were traditionally thought to involve smooth ER, i.e., membrane regions of the ER devoid of ribosomes, and represent close contacts (∼10–15 nm) between the two organelles (Goetz and Nabi, 2006); ribosome-studded rough ER riboMERCs (25-nm contact distance) are found in liver and called WrappER as they wrap around mitochondria (Anastasia et al., 2021; Csordás et al., 2006; Giacomello and Pellegrini, 2016; Ilacqua et al., 2022). Wider (50–60 nm)

riboMERCs were identified in metastatic HT-1080 fibrosarcoma cells and in HEK293 cells where they are regulated by the Gp78 E3 ubiquitin ligase and interaction between mitochondrial outer membrane protein 25 (OMP25), also called Synaptojanin-2-binding protein (SYNJ2BP), and its ER partner, ribosome-binding protein 1 (RRBP1), respectively (Hung et al., 2017; Wang et al., 2015). Other studies have reported MERCs of varying distances ranging from 10 to 80 nm (Giacomello and Pellegrini, 2016), highlighting the diversity of MERCs.

The specificity of labeling and ability to study dynamic fluorescent-tagged proteins in living cells makes fluorescent microscopy the method of choice to characterize the diversity, dynamics, and molecular mechanisms underlying MERC formation. However, analysis of MERCs by fluorescence microscopy faces three major hurdles: (1) the distance between the ER and mitochondria is below the resolution of optical microscopy (200–250 nm) due to diffraction limits; (2) MERC segmentation approaches are sensitive to subjective parameter settings such that accurate thresholding of the ER, ranging from isolated peripheral tubules to the dense central ER matrix, is particularly

[1]School of Computing Science, Simon Fraser University, Burnaby, Canada; [2]Department of Cellular and Physiological Sciences, Life Sciences Institute, University of British Columbia, Vancouver, Canada; [3]School of Biomedical Engineering, University of British Columbia, Vancouver, Canada.

*B. Cardoen, K.R. Vandevoorde, G. Gao, and M. Ortiz-Silva contributed equally to this paper;   Correspondence to Ivan R. Nabi: irnabi@mail.ubc.ca;   Ghassan Hamarneh: hamarneh@sfu.ca

**I.R. Nabi and G. Hamarneh are joint senior authors.

challenging; (3) bifluorescent complementation systems of varying linker lengths present differential detection of MERCs, but may promote or stabilize MERC formation (Cieri et al., 2018; Harmon et al., 2017; Vallese et al., 2020). Earlier work using diffraction-limited confocal microscopy showing that ER tubules mark mitochondrial constrictions (Friedman et al., 2011) has been confirmed using super-resolution single-molecule localization microscopy (SMLM; Shim et al., 2012) and live cell 2D stimulated emission depletion (STED) imaging (Bottanelli et al., 2016). 2D STED characterized roles for ER shaping proteins in control of peripheral ER tubule nanodomains and fenestrations in ER sheets (Gao et al., 2019; Schroeder et al., 2019). 3D super-resolution whole-cell analysis by structured illumination (SIM) or 3D STED microscopy achieves ~120 nm lateral and ~250 nm axial resolution and identified tubular matrices in peripheral ER sheets and the dense central ER of Zika virus–infected cells (Long et al., 2020; Nixon-Abell et al., 2016). SIM and STED therefore represent optimal super-resolution imaging approaches to study the distribution and morphology of MERCs in 3D whole-cell views.

However, the intervening space between ER and mitochondria remains far smaller than the resolution provided by SIM or 3D STED. In addition, detection of MERCs requires analytical approaches that can accurately assess overlap between two fluorescent channels acquired independently, facing challenges of varying background density and difficulty of thresholding signals of varying signal-to-noise ratio (SNR). Recently, the optimal transport distance between two fluorescence distributions was formulated as an alternative approach to colocalization (Tameling et al., 2021; Wang and Yuan, 2021). However, whether this method is able to recover interactions at subprecision distances of MERCs is not known. By reducing the anisotropy using multiple unregistered samples and mathematical optimization, more precise colocalization is becoming possible (Fortun et al., 2018); however, this method does not directly quantify subprecision interfaces. Quantitative detection of MERCs in 3D cell volumes for fluorescent microscopy therefore requires novel analytical approaches that apply subpixel resolution detection of interaction zones to whole-cell super-resolution imaging approaches.

To this end, we developed a multichannel differential analysis to reconstruct the interface at subpixel precision: MCS-DETECT. Without segmentation, the algorithm adapts robustly to the intensity variations between fluorescent channels and samples, resulting in highly sensitive detection independent of variations in local signal or background intensity differentials. Application to 3D STED super-resolution microscopy whole-cell image volumes distinguishes different classes of MERCs and defines novel dual control of the formation of a distinct class of extended, convoluted, tubular riboMERCs. Formation of tubular ribo-MERCS is dependent on expression of the riboMERC tether RRBP1 and their size and shape modulated by Gp78 E3 ubiquitin ligase activity.

## Results

### Distinct MERCs in COS-7 and HT-1080 cells
EM analysis shows the presence of elongated rough ER–mitochondria (RER-mito) contacts in HT-1080 cells, as previously reported (Wang et al., 2015), while COS-7 cells present predominantly shorter, smooth ER–mitochondria (SER-mito) contacts (Fig. 1). Further, SER-mito and RER-mito contacts coexist in HT-1080 cells as a single unit, with a smaller smooth MERC extending from a more elongated RER-mito contact (Fig. 1 A), as previously reported (Giacomello and Pellegrini, 2016; Wang et al., 2015). Defining RER-mito contacts as any contact site containing at least one ribosome within the interorganellar space, we report here a 55-nm spacing between ER and mitochondria of HT-1080 RER-mito contacts and 15-nm spacing of SER-mito contacts, corresponding to our previous study (Wang et al., 2015). In COS-7 cells, RER-mito contacts are ~35 nm in width while SER-mito contacts are ~25 nm in width (Fig. 1 B). The different spacing of both classes of MERCs in HT-1080 and COS-7 cells highlights the varied spacing of MERCs in different cells and tissues (Csordás et al., 2006; Giacomello and Pellegrini, 2016; Hung et al., 2017; Wang et al., 2015).

RER-mito contacts in HT-1080 are of varying length and significantly longer than RER-mito contacts in COS-7 cells or SER-mito contacts in either HT-1080 or COS-7 cells (Fig. 1 B). HT-1080 RER-mito contacts extend over almost 30% of the mitochondrial perimeter and contact a significantly larger proportion of mitochondria analyzed compared with SER-mito contacts or COS-7 RER-mito contacts that contact <5% of the mitochondrial surface. Almost 70% of HT-1080 MERCs are RER-mito contacts; by length, RER-mito contacts compose 90% of MERCs in HT-1080 cells compared to <10% of MERCs in COS-7 cells (Fig. 1 C). Importantly, RER-mito contacts of COS-7 cells contained at most five ribosomes within the interorganellar space compared with HT-1080 RER-mito contacts, some of which contained up to 25 ribosomes in a single contact (Fig. 1 D). Elongated RER-mito contacts containing >5 ribosomes are therefore specific to HT-1080 cells and will heretofore be referred to as riboMERCs (Giacomello and Pellegrini, 2016).

Whole-cell 3D STED image stacks of HT-1080 and COS-7 cells transfected with the lumenal ER reporter, ERmoxGFP (Costantini et al., 2015), were fixed with ER-preserving 3% paraformaldehyde/0.2% glutaraldehyde (Gao et al., 2019; Long et al., 2020; Nixon-Abell et al., 2016) and labeled for the outer mitochondrial membrane (OMM) protein TOM20 (Fig. 2 A). Overlapping ER–mitochondria signal is observed in the cell periphery and more extensively in the central ER region, which for HT-1080 cells extends across multiple STED sections over 3 microns in depth. COS-7 cells present an abundance of ER–mitochondria overlapping regions but limited central ER overlap with mitochondria (Fig. 2 A). The extended ER–mitochondria overlap of HT-1080 cells is consistent with the presence of elongated riboMERCs, as observed by EM (Fig. 1). However, defining actual contact sites from these 3D fluorescent whole-cell views is challenging and subjective, based on user-dependent segmentation.

### MERC identification by differential channel correlation
Current image segmentation in 3D STED faces challenges of varying background density leading to difficulty of thresholding signals of varying density, while ground truth annotation is unavailable. This problem is particularly challenging when

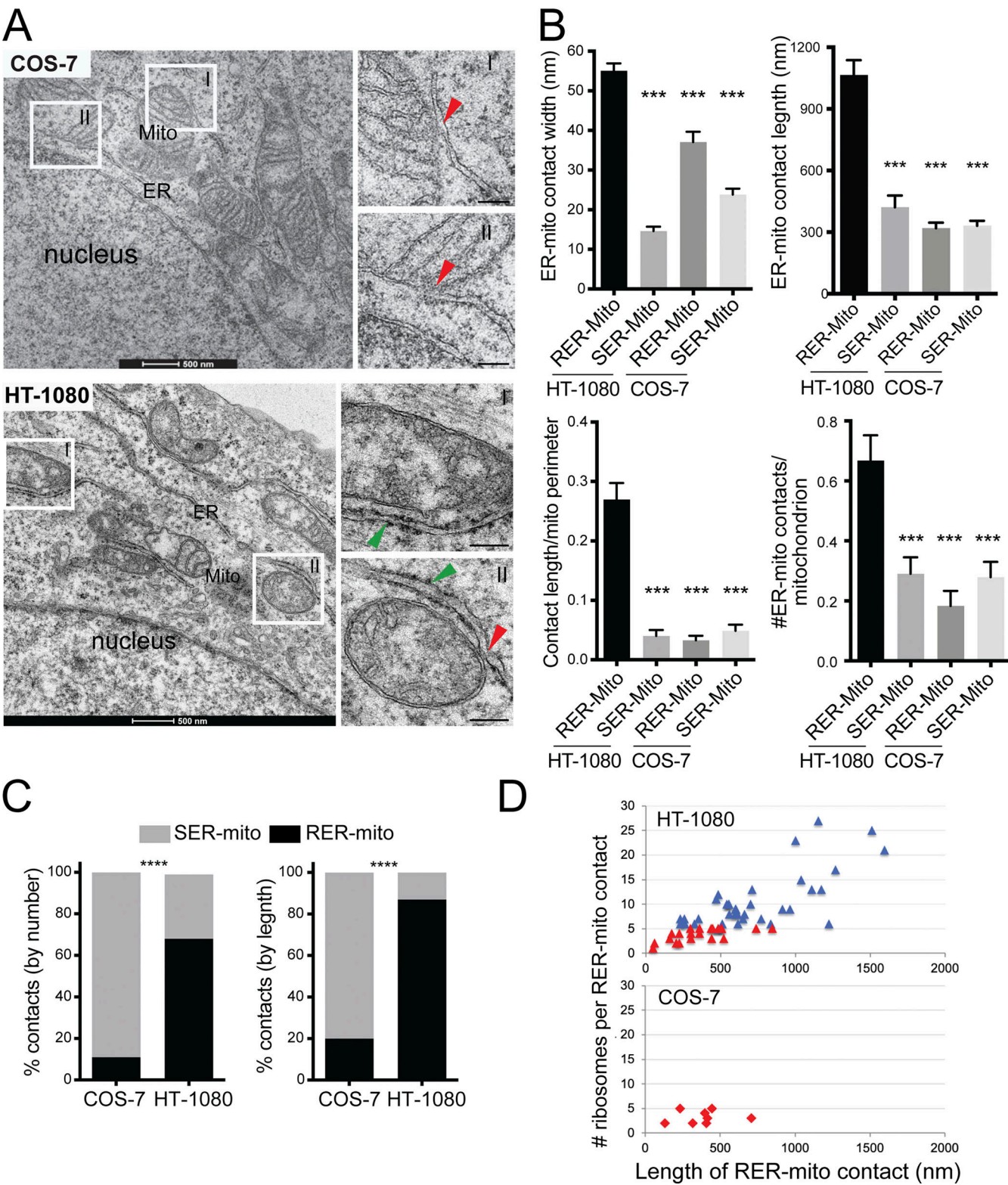

**Figure 1.** **Quantitative EM analysis of ER–mitochondria contacts in HT-1080 and COS-7 cells. (A)** Representative EM images of HT-1080 and COS-7 cells. Insets show rough ER–mitochondria contacts (RER-mito) in HT-1080 cells (green arrowheads) and smooth ER–mitochondria contacts (SER-mito) in HT-1080 and COS-7 cells (red arrowheads). **(B)** Quantification of contact width, contact length, contact length relative to mitochondria perimeter, and number of contacts per mitochondria profile are shown for SER-mito and RER-mito contacts in HT1080 and COS-7 cells. **(C)** The relative ratio of SER-mito and RER-mito contacts in HT-1080 and COS-7 cells based on the number of contacts per mitochondria or length of contacts. **(D)** The number of ribosomes per RER-mito contact is plotted versus the length of the contact in nm for HT-1080 and COS-7 cells. RER-mito contacts with five or less ribosomes are shown in red; those with more than five ribosomes are specific to HT-1080 cells and are shown in blue and defined as riboMERCs. $n = 27$ images from two independent biological replicates; ±SEM; ***P < 0.001; ****P < 0.0001; B: one-way ANOVA; C: Chi2 test. Bar = 500 nm; inset: 200 nm.

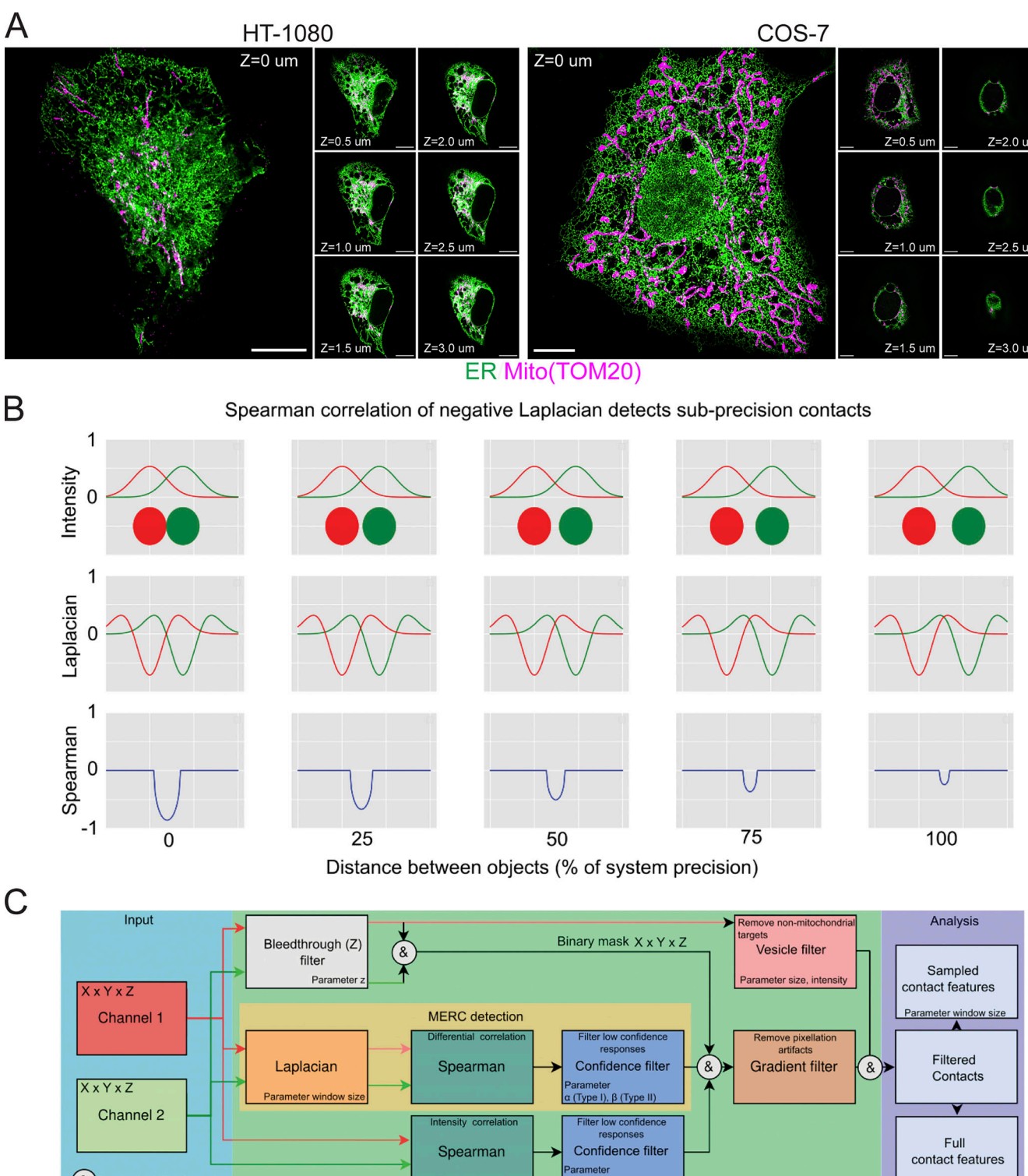

Figure 2. **MCS-DETECT analysis of sub-precision contacts. (A)** 3D STED images of HT-1080 and COS-7 showing overlap between mitochondria (magenta) and ER (green). Insets show STED sections at 0.5 µm Z spacing. Bars = 10 µm. **(B)** Two objects (red and green discs) are shown at corresponding sub-precision distances. Intensity profiles (top row), second derivatives (Laplacian), and Spearman correlations of the negative part of the Laplacian (bottom row) are shown. Note how the Spearman response overlaps and changes consistently with the sub-precision distance. **(C)** The detection algorithm (orange) with additional stages that each address a specific confounding factor introduced by the acquisition (bleed through) or sample (vesicle removal).

dealing with the highly varied density and complexity of the ER. In addition, analysis of overlap between two channels is based on separate segmentation of independent channels, compounding the error in capturing interaction. Here, we detect MERCs, below precision of the microscope, in this case, 3D STED, by observing how the relative intensity of ER and mitochondria labeling change in tandem. Images are scanned using a sliding window applying a differential operator to approximate local signal differentials. Interaction between channels colocalizes with the negative Spearman correlation of the image differential (Fig. 2 B). Contact zones are detected without requiring segmentation representing a novel approach to detect organelle contact sites from multichannel fluorescent images.

An in silico experiment demonstrates the underlying principle of the method (Fig. 2 B). An axial view of the intensity of two spheres acquired with a Gaussian point spread function (PSF) is shown in the upper row and negative Laplacian of the intensity profile of the two objects in the middle row. Distance between the two objects is varied from direct interaction up to the system resolution to mimic the MERC reconstruction problem where the interface is below acquisition precision. The detection principle relies on the negative correlation (bottom row, Fig. 2 B) of the second intensity differential, approximated by the Laplacian operator. Importantly, the differential Spearman response is present for the entire subprecision interaction range, confirming that a negative Spearman correlation of the negative Laplacian of the intensity profile corresponds to overlap of two adjacent objects, even when the precision of the system does not allow direct observation. The geometric mean of the Spearman correlation is reported for each contact. While the Spearman correlation can be impacted by variation in size and shape of the contact, at the cell level, we observe consistent patterns across cell lines. As shown in Fig. 2 C, the correlation principle is one step of a larger sequence of steps that address variable labeling density, low SNR, intensity bleeding through multiple Z-slices, and anisotropic precision associated with 3D STED images. A full parameter sensitivity study is performed, with results shown in Fig. S1 A. The full algorithm, including pseudocode and mathematical formulation for each stage, is detailed in Materials and methods and Fig. S1 B.

### Detecting riboMERCs in HT-1080 cells with MCS-DETECT
Application of MCS-DETECT to HT-1080 and COS-7 cells transfected with ERmoxGFP and labeled for mitochondrial TOM20 extracts a mask of contact zones (Fig. 3 A). Overlay of the detected contacts on 3D volume rendering of deconvolved STED ER and mitochondria image stacks shows extended perinuclear contact zones in HT-1080 cells and smaller dispersed MERCs along mitochondria in COS-7 cells. Insets show that contact sites are localized to regions of interaction between ER (in green) and mitochondria (in magenta). 3D STED super-resolution microscopy is anisotropic with a predicted lateral resolution of 120 nm and axial resolution of 250 nm. Each voxel is 25 × 25 × 100 nm such that contact zone detection for larger contacts, particularly for those that are located on top of mitochondria parallel to the plane of the acquired optical section, may be recovered with lower precision. We also detected a large

number of contact sites between ER and smaller, reduced-intensity TOM20-labeled mitochondrial structures (Fig. S2), which may correspond to mitochondria vesicles (Neuspiel et al., 2008). To ensure that MERC analysis by MCS-DETECT parallels the contact sites detected by EM adjacent to intact mitochondria (Fig. 1), we filtered out these small, low-intensity TOM20-labeled ER-associated structures from the images based on size and intensity (Fig. S2).

To quantify ER–mitochondria contact zones in HT-1080 and COS-7 cells, we detected, per cell, mitochondria surface coverage ratio and number of mitochondria contacts within a fixed, non-overlapping, sliding window over the surface of each mitochondrion (Fig. 3 B). HT-1080 cells present an approximate twofold increased coverage of mitochondria and number of contacts per sample region, corresponding closely to the increased mitochondrial surface coverage we observe by EM (Fig. 1 B), if one combines the smooth and riboMERC EM data as these are indistinguishable by STED microscopy. Pairwise analysis of MERC features shows that HT-1080 MERCs are on average larger and more spherical than COS-7 MERCs (Fig. 3 C). A clear difference in anisotropy and mean Spearman response supports structural differences between MERCs of HT-1080 and COS-7 cells that show a clear correspondence with our EM results (Fig. 1).

The OMM AKAP1 (A kinase anchor protein) sequence fused to the ER targeting sequence of Ubc is an OMM–ER linker that induces close contacts (<20 nm) between ER and mitochondria (Csordás et al., 2010; Hajnóczky et al., 2006). Expression in COS-7 cells of this construct induced extended contact sites (Fig. 3 A). Mitochondrial surface coverage was increased relative to COS-7 cells but decreased relative to HT-1080 cells. In contrast to the MERCs of HT-1080 cells, a reduction in the number of contact sites per sample region was observed. As for HT-1080 cells, pairwise analysis showed that the MERCs of COS-7 cells expressing the OMM–ER linker were larger and showed distinct groupings compared with COS-7 MERCs with respect to both sphericity and Spearman response. The latter features were not, however, shifted away from COS-7 to the same extent as HT-1080 MERCs. The OMM–ER linker therefore induces expansion of COS-7 MERCs to form larger MERCs whose features are distinct from riboMERC expressing HT-1080 cells. MCS-DETECT therefore detects MERC features that form a statistically significant signature for diverse types of contacts.

### Gp78 and RRBP1 are independent regulators of riboMERC expression
Using CRISPR/Cas knockout (KO) Gp78 HT-1080 cells that present deficient basal mitophagy, impaired mitochondrial health, and increased mitochondrial ROS (Alan et al., 2022), we show that loss of Gp78 reduces MERC mitochondrial coverage (Fig. 4 A). Surface coverage and number of contacts per mitochondrial surface region are significantly reduced relative to HT-1080 cells but remain elevated compared with COS-7 cells (Fig. 4 B). The partial reduction of riboMERCs upon Gp78 KO is consistent with prior EM results showing that Gp78 knockdown does not completely eliminate riboMERCs (Wang et al., 2015). Overexpression of Gp78 in COS-7 cells induces the inverse

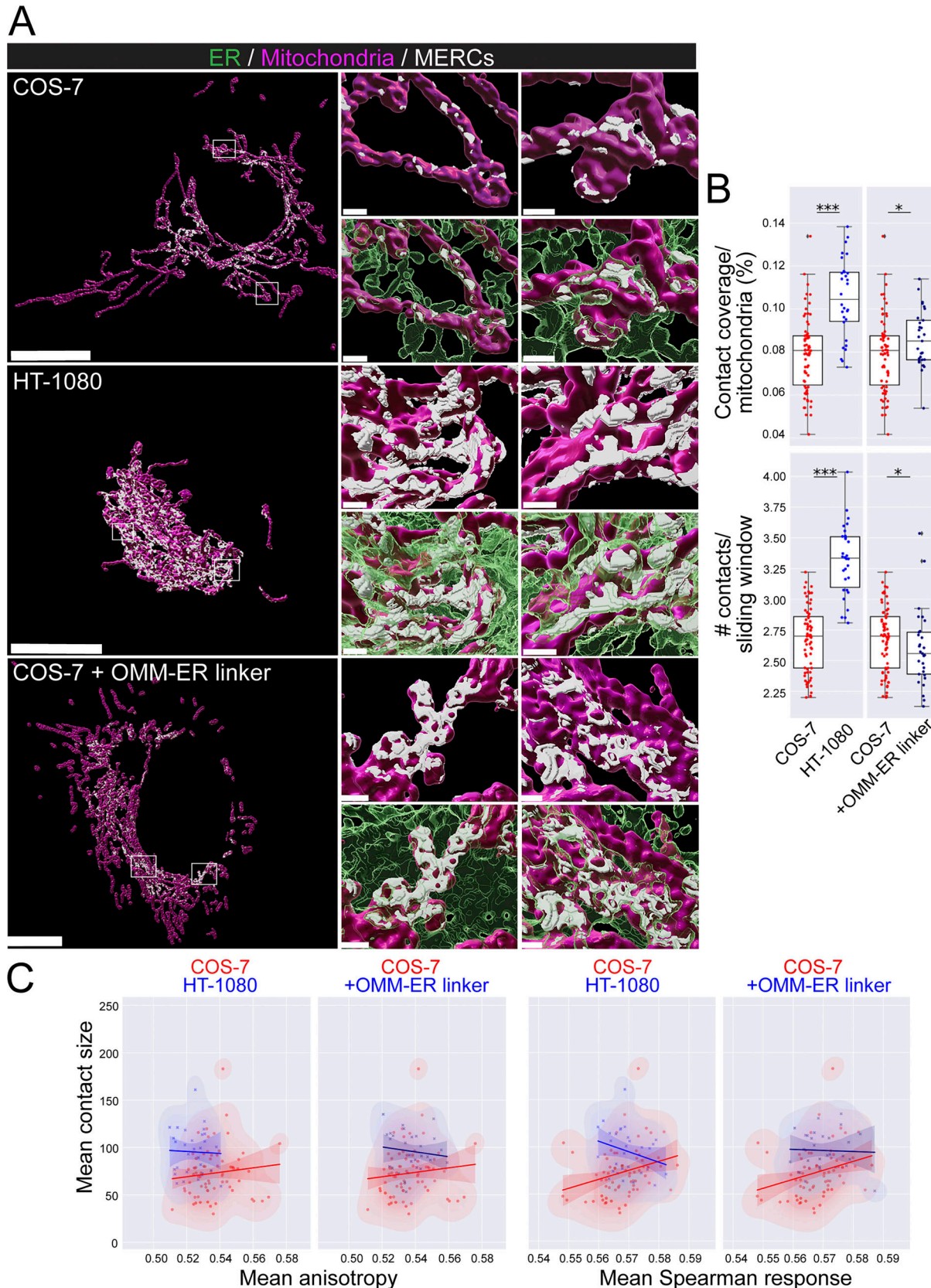

Figure 3. **Subprecision contact detection identifies distinct contact profiles in HT-1080 and COS-7 cells. (A)** Volume-rendered MCS-DETECT views of cells expressing ERmoxGFP (green) and labeled for TOM20 (magenta) with contact sites overlaid (white) are shown for COS-7, HT-1080, and OMM–ER linker transfected COS-7 cell ROIs from the whole view image are shown volume rendered in adjacent panels. COS-7 mitochondria display numerous small contact zones while mitochondria in HT-1080 and OMM–ER linker transfected COS-7 cells present more extended contact zones (bar = 10 μm whole cell; 1 μm insets).

**(B)** Mitochondria surface coverage ratio and the number of contacts per sampled mitochondria window are shown for contact zones in COS-7, HT-1080, and OMM–ER linker transfected COS-7 cells (averaged over cell, two-sided non-parametric Mann Whitney test, n = 3 independent biological replicates, ≥30 cells/condition per experiment; *P < 0.05; ***P < 0.001). **(C)** 2D KDE plots of mean contact size over mean anisotropy and mean Spearman response, with a linear regression overlayed, are shown for COS-7 (red) versus HT-1080 (blue) cells or COS-7 (red) versus OMM–ER linker transfected COS-7 (blue) cells.

response in which COS-7 cells gain large contacts similar to those found in HT-1080 cells (Fig. 4, A and B; and Fig. S3 A). Importantly, overexpression of a Ring finger mutant of Gp78, Gp78 RM, lacking ubiquitin ligase activity and unable to induce mitophagy (Fang et al., 2001; Fu et al., 2013), does not induce this effect. Pairwise feature analysis shows that Gp78 expression in COS-7 cells induces size-dependent increase in anisotropy and reduction of mean Spearman response that matches those observed in HT-1080 cells (Fig. 4 C). To determine if Gp78-dependent formation of riboMERCs had a functional impact, we measured mitochondrial potential of the transfected cells using the potential-dependent mitochondrial fluorescent reporter MitoView633 (Alan et al., 2022). Gp78 enhanced mitochondrial potential of COS-7 cells while expression of the Gp78 Ring finger mutant and the OMM–ER linker reduced mitochondrial potential (Fig. 4 D and Fig. S3 B). These data indicate that the large contact zones selectively enriched in HT-1080 cells correspond to riboMERCs and identify Gp78 ubiquitin ligase activity as a specific regulator of the size of these contact sites.

A parallel analysis was then undertaken in HeLa cells (Fig. 5). Overexpression of Gp78 increased mitochondria surface coverage and number of contacts per sliding window, although to a lesser extent than in COS-7 cells (Fig. 3). The Gp78 Ring finger mutant increased contact coverage to a lesser extent than wild-type (WT) Gp78 and did not significantly impact the number of contacts. MCS-DETECT detected a significant increase in number of contacts for the OMM–ER linker but not an increase in mitochondrial surface coverage (Fig. 5). The reduction in mitochondrial surface coverage induced by Gp78 and the OMM–ER linker in HeLa cells relative to COS-7 is likely related to increased expression levels upon transfection in COS-7 cells (Fig. S4 A). Expression of the Gp78 Ring finger mutant was not increased in COS-7 cells, perhaps due to cytotoxicity at high expression levels. Mitochondrial potential of HeLa cells was increased by expression of Gp78 but not impacted by expression of the Gp78 Ring finger mutant or the OMM–ER linker (Fig. 5 C and Fig. S3 C). These data show that Gp78 ubiquitin ligase activity is able to induce large riboMERCs in two distinct cell lines.

Previous studies identified RRBP1 as a MERC-resident protein required for increased riboMERC expression through interaction with its mitochondrial partner SYNJ2BP (Hung et al., 2017). To investigate the impact of RRBP1 on riboMERC formation in HT-1080 cells, WT and Gp78 KO HT-1080 cells were treated with either control siRNA or siRNA targeted to RRBP1 and analyzed by EM (Fig. 6). WT HT-1080 cells displayed both the highest number of riboMERCs per mitochondria as well as the highest ratio of riboMERC length to mitochondrial perimeter, which showed a partial reduction in Gp78 KO cells, consistent with our MCS-DETECT analysis (Fig. 4) and prior EM studies of Gp78 shRNA knockdown HT-1080 cells (Wang et al., 2015). However, upon siRNA knockdown of RRBP1 (Fig. S4 B),

we observed an almost complete ablation of riboMERCs (Fig. 6 B). Analysis of MERC width revealed that the few remaining riboMERCs in siRRBP1-treated Gp78 KO cells had a slight if significantly larger width (Fig. 6 C), perhaps reflecting increased spacing of riboMERCs as observed in siRRBP1-treated hepatocytes (Anastasia et al., 2021). MCS-DETECT analysis of 3D STED super-resolution images of siRRBP1-transfected cells mirrored the EM analysis; RRBP1 knockdown decreased both the absolute number of contacts as well as the percentage of the mitochondrial surface covered by MERCs, irrespective of Gp78 expression (Fig. 7, A and B). These values in siRRPB1-treated HT-1080 cells were similar to those observed in COS-7 cells (Fig. 4), which also show an absence of riboMERCs. Importantly, siRRBP1 knockdown reduced mitochondrial potential in both WT and Gp78 KO HT-1080 cells, highlighting the importance of riboMERCs to maintenance of mitochondrial potential (Fig. 7 C and Fig. S3 D). Similarities in MCS-DETECT reporting on MERC coverage in COS-7 and HeLa cell lines, in which riboMERCs are induced by Gp78 expression, and in HT-1080, in which they are lost by siRRBP1 knockdown, highlight the robustness of the MERC detection approach.

## Tubular, Gp78-dependent riboMERCs

To characterize riboMERCs at the level of individual MERCs, we analyzed the largest 5% of MERCs in each cell, which, based on our EM analysis, corresponds to elongated riboMERCs in HT-1080 cells (Fig. 1). Analysis of the size of the largest 5% of MERCs per cell (Volume Q95) in the various cells and treatments analyzed (Fig. 8 A) paralleled our analysis of the average size of all MERCs per cell (Figs. 3 B and 4 B). This represents a strong indication that differences in contacts across cell lines are driven by changes in the tail (largest) of the contact size distribution. The highly significant differences between the largest 5% MERCs between HT-1080 and both COS-7 and siRRBP1-treated HT-1080 cells indicate that the largest 5% of HT-1080 MERCs encompass predominantly riboMERCs. Overexpression of Gp78 in COS-7 and HeLa cells increased the size of top 5% MERCs, with Gp78-induced Q95 MERCs in COS-7 cells even larger than those of WT HT-1080 cells. We further counted the number of large MERCs, larger than the 500 voxel average size of the largest 5% of HT-1080 MERCs (Fig. 8 A). The number of large MERCs was significantly reduced in COS-7 relative to HT-1080 cells and upon siRRBP1 knockdown in HT-1080, consistent with the absence of riboMERCs in those cells (Figs. 1 and 6). The number of large MERCs was increased in COS-7 cells upon expression of the OMM–ER linker, although not in HeLa cells, and not to the extent observed upon expression of Gp78. We attribute the reduced size of large Gp78-induced MERCs and lack of an effect on large MERCs of the OMM–ER linker to reduced expression of these constructs when expressed in HeLa relative to COS-7 cells (Fig. S3 C). Interestingly, the top 5% of MERCs in

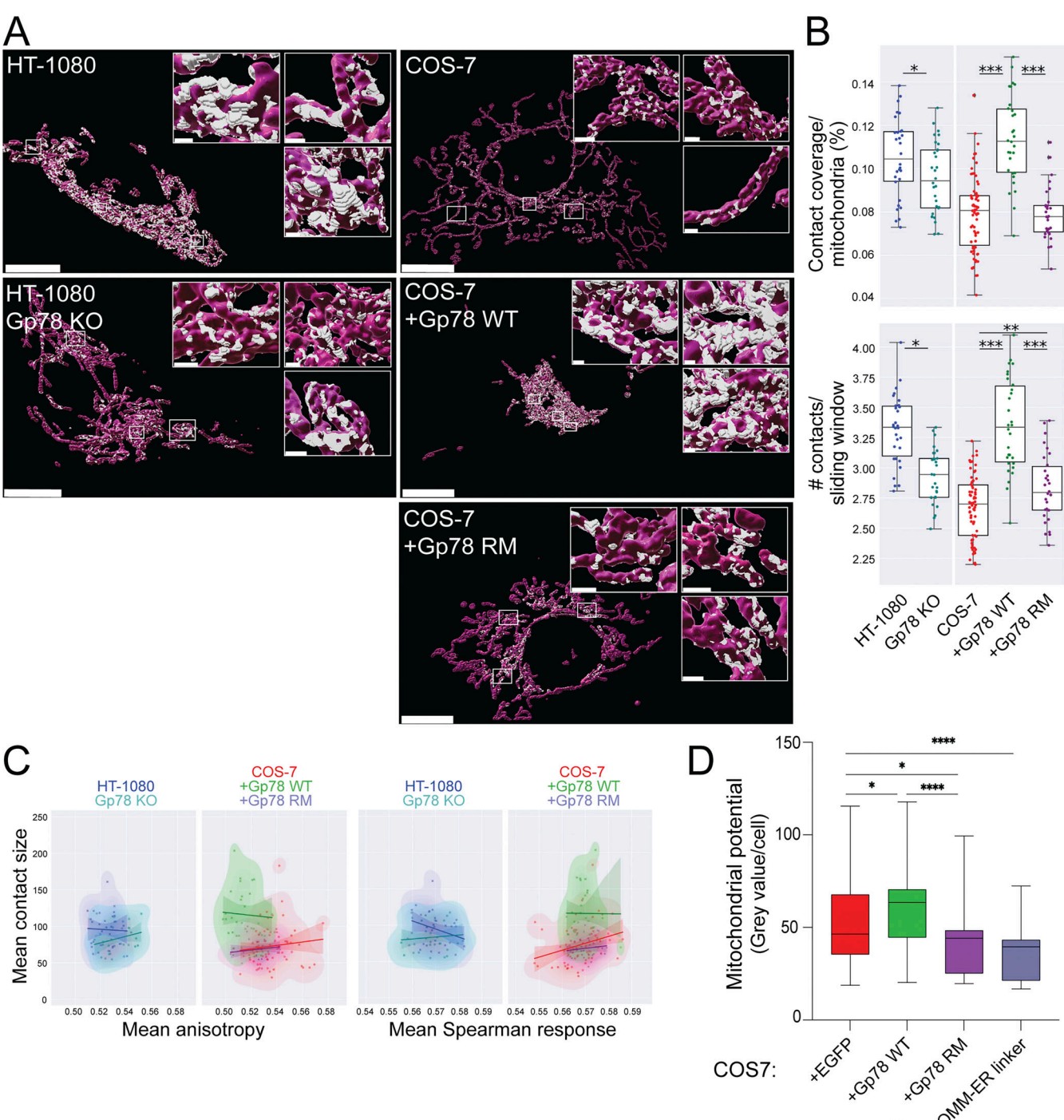

Figure 4. **Gp78 regulation of riboMERCs. (A)** Volume-rendered MCS-DETECT views of cells expressing ERmoxGFP and labeled for TOM20 (magenta) with contact sites overlaid (white) are shown for HT-1080 and Gp78 KO HT-1080 cells and for untransfected COS-7 cells and COS-7 cells overexpressing WT Gp78 or Gp78 RM. Bar = 10 µm whole cell; 1 µm insets. **(B)** Mitochondria surface coverage ratio and the number of contacts per sampled mitochondria window are shown for contact zones in HT-1080 and Gp78 KO HT-1080 cells and for untransfected COS-7 cells and COS-7 cells overexpressing Gp78 WT or Gp78 RM. **(C)** 2D KDE plots of mean contact size over mean anisotropy and mean Spearman response, with a linear regression overlayed, are shown for HT-1080 (blue) versus Gp78 KO HT-1080 (green) cells or COS-7 (red) versus COS-7 overexpressing either Gp78 WT (green) or Gp78 RM (blue). Averaged over cell, $n = 3$ independent biological replicates, ≥30 cells/condition per experiment; *$P < 0.05$; **$P < 0.01$; ***$P < 0.001$, two-sided non-parametric Mann–Whitney test. **(D)** COS-7 cells were transfected with EGFP (as a control), Gp78 WT IRES-GFP, Gp78 RM IRES-GFP, or the OMM–ER linker (RFP) and labeled with MitoView 633. Integrated density of MitoView 633 per cell was quantified. $n = 3$ independent biological replicates; >35 cells/condition per experiment; *$P < 0.05$; ****$P < 0.0001$; Tukey post hoc test.

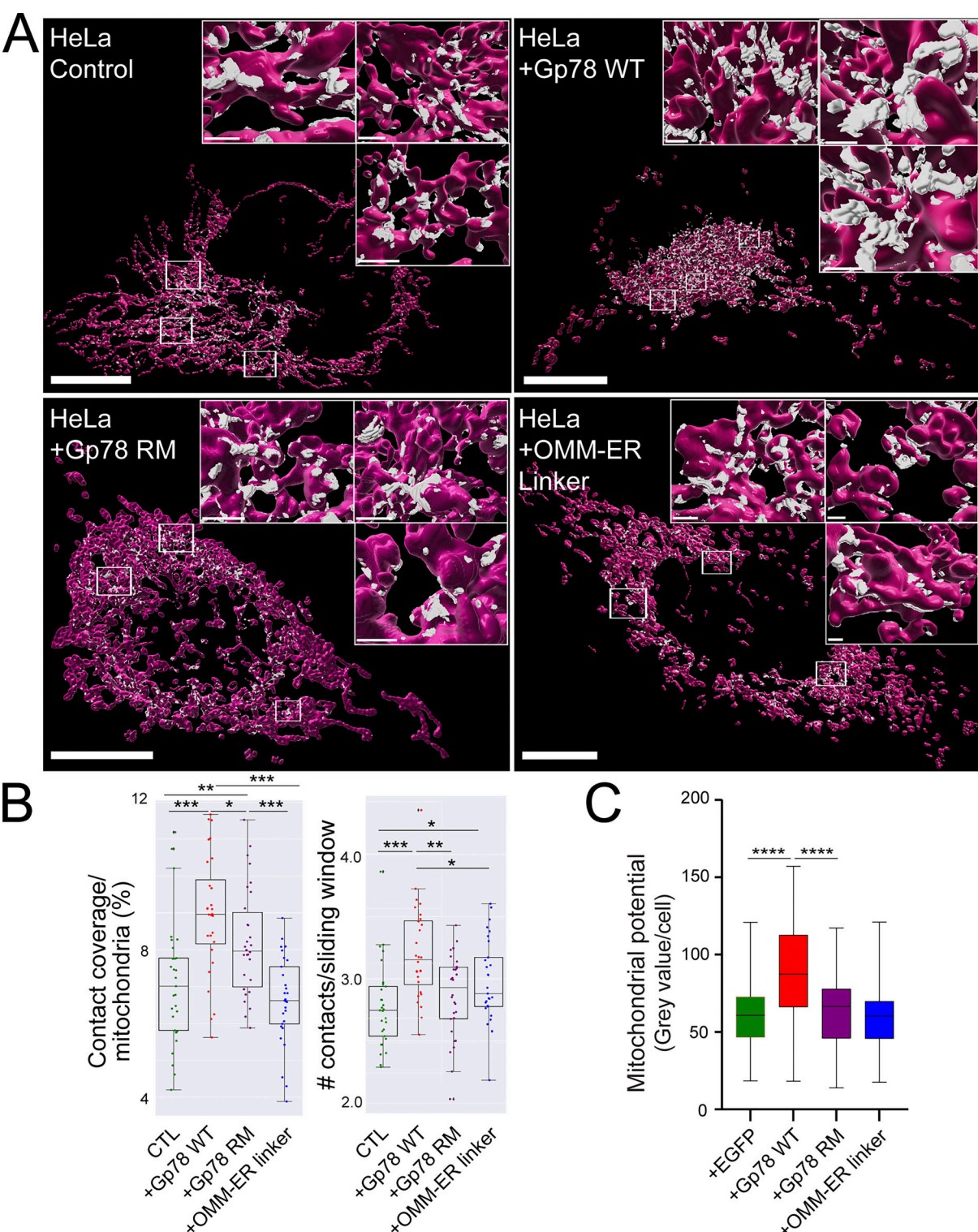

Figure 5. **Gp78 induces riboMERCs in HeLa cells. (A)** Volume-rendered MCS-DETECT views of cells expressing ERmoxGFP and labeled for TOM20 (magenta) with contact sites overlaid (white) are shown for untransfected HeLa cells and HeLa cells overexpressing WT Gp78, Gp78 RM, and the OMM–ER linker. Bar = 10 µm whole cell; 1 µm insets). **(B)** Mitochondria surface coverage ratio and the number of contacts per sampled mitochondria window are shown for contact zones in the cells indicated above. Averaged over cell, two-sided non-parametric Mann–Whitney test, $n$ = 3 independent biological replicates, ≥30 cells/ condition per experiment; *P < 0.05; **P < 0.01; ***P < 0.001. **(C)** HeLa cells were transfected with EGFP (as a control), Gp78 WT, Gp78 RM, or the OMM–ER linker and labeled with MitoView 633. Integrated density of MitoView 633 per cell was quantified. $n$ = 3 independent biological replicates; >35 cells/condition per experiment; ****P < 0.0001; Tukey post hoc test.

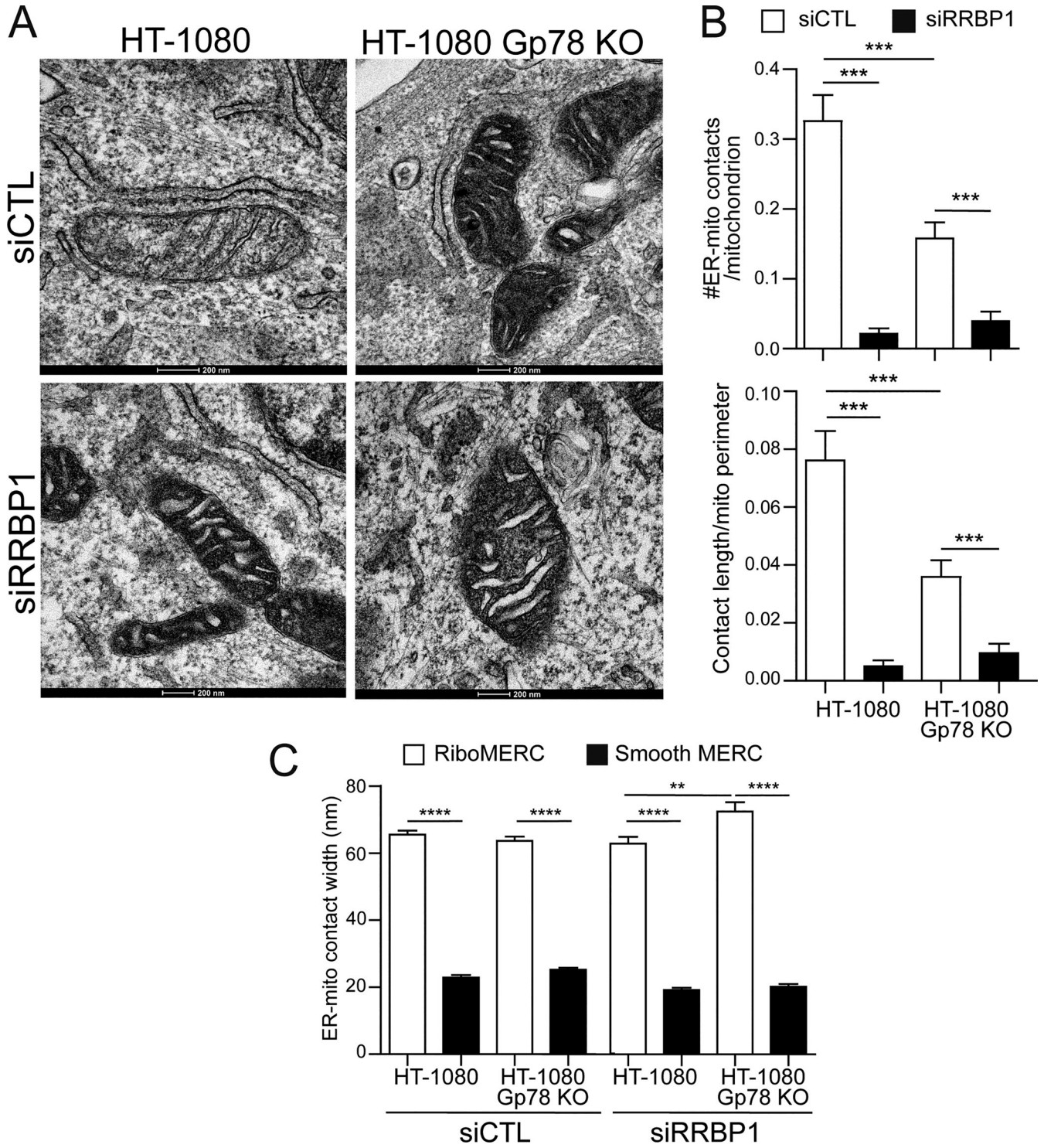

Figure 6. **RRBP1 knockdown reduces riboMERCS independent of Gp78. (A)** Representative EM images of HT-1080 and HT-1080 Gp78 KO cells treated with either siControl or siRRBP1. Images highlight the presence of riboMERCS in both HT-1080 WT and Gp78 KO cells, which are almost completely lost upon RRBP1 knockdown. **(B)** Quantification of the number of riboMERCs per mitochondria and the ratio of riboMERC length to mitochondrial perimeter for the conditions in A. **(C)** Quantification of the MERC width for both riboMERCs and smooth MERCs for the conditions in A. $n = 31$ images from two independent biological replicates; **$P < 0.01$; ***$P < 0.001$; ****$P < 0.0001$; unpaired $t$ test. Bar = 200 nm.

Gp78 KO HT-1080 cells were smaller than in HT-1080 cells, yet the number of large (>500 voxel) MERCs was the same in the two cell lines, suggesting that Gp78 KO selectively regulates the size of riboMERCs as opposed to their abundance. Together, these EM and MCS-DETECT data demonstrate that, in contrast to RRBP1, which is required for riboMERC expression in HT-1080 cells, Gp78 rather acts to modulate the size of riboMERCs via its ubiquitin ligase activity.

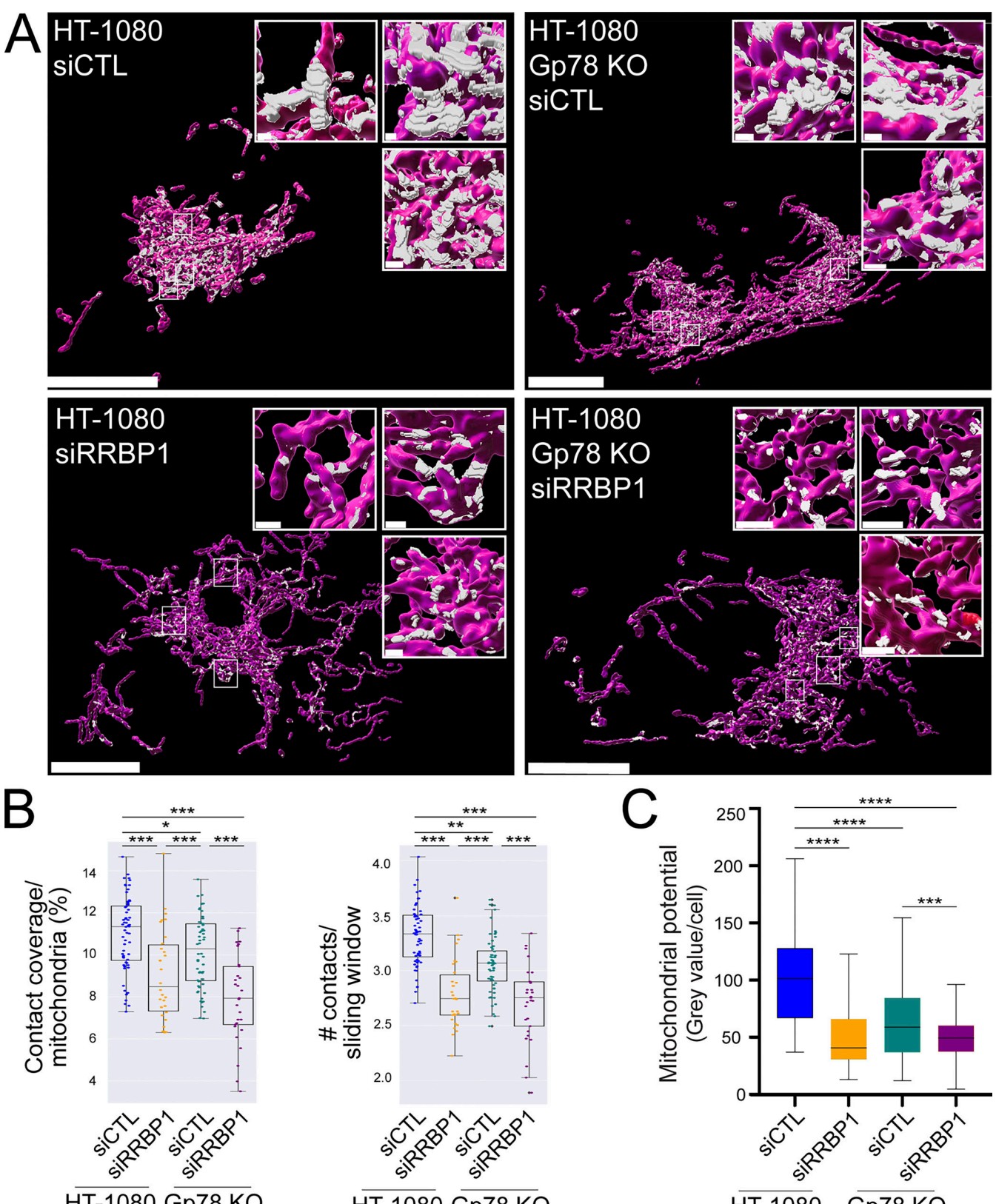

Figure 7. **MCS-DETECT captures MERC changes induced by RRBP1 knockdown. (A)** Volume-rendered MCS-DETECT views of HT-1080 WT and Gp78 KO cells treated with either siControl or siRRBP1. Mitochondria are labeled with TOMM20 (red) and MERCS are visualized in white. Bar = 10 μm whole cell; 1 μm insets. **(B)** Mitochondria surface coverage ratio and the number of contacts per sampled mitochondria window are shown for contact zones in HT-1080 WT and Gp78 KO cells treated with either siControl or siRRBP1. Averaged over cell, two-sided non-parametric Mann–Whitney test, $n$ = 3 independent biological replicates, ≥30 cells/condition per experiment; *P < 0.05; **P < 0.01; ***P < 0.001. **(C)** HT-1080 cells transfected with either siControl or siRRBP1 were labeled with MitoView633. Integrated density of MitoView633 per cell was quantified. $n$ = 3 independent biological replicates; >50 cells/condition per experiment; ***P < 0.0001; ****P < 0.0001; Tukey post hoc test.

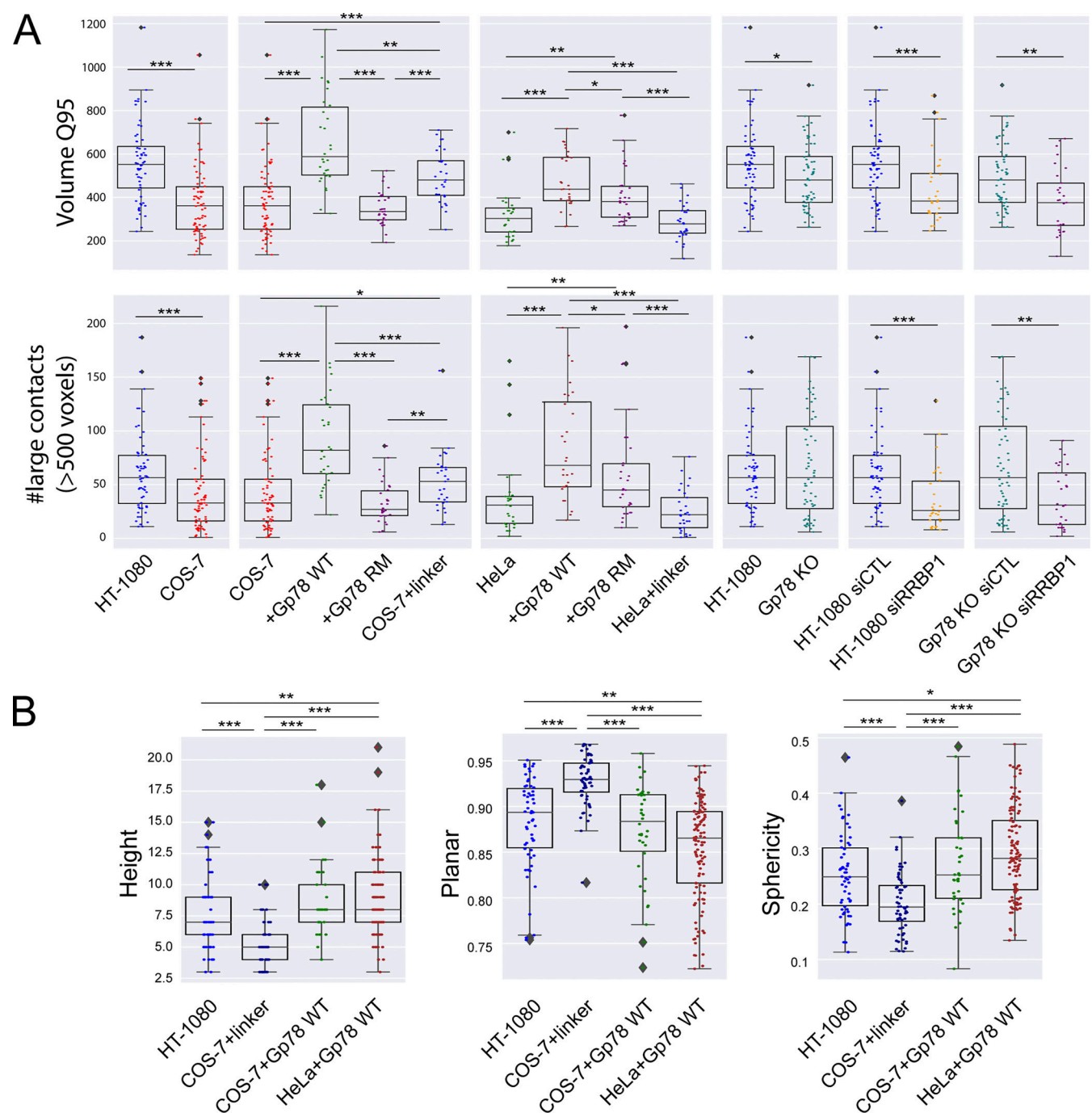

Figure 8. **Large MERCs induced by Gp78 and the OMM–ER linker present distinct shape signatures. (A)** The 95th quantile of MERC volume per cell (Q95V; largest 5% of MERCs per cell) and number of MERCs per cell larger than the average 500-voxel size of HT-1080 Q95V MERCs are shown for HT-1080 and COS-7 cells, COS-7, and COS-7 cells overexpressing either Gp78 WT, Gp78 RM, or the OMM–ER linker, HeLa and HeLa cells overexpressing either Gp78 WT, Gp78 RM, or the OMM–ER linker, HT-1080, and Gp78 KO HT-1080 cells, HT-1080 cells transfected with siCTL and siRRBP1, and Gp78 KO HT-1080 cells transfected with siCTL and siRRBP1. **(B)** Representative cells whose Q95V is closest to the mean Q95V for HT-1080 cells, for COS-7 or HeLa cells over-expressing Gp78 WT, and for COS-7 cells overexpressing the OMM–ER linker were selected for analysis. For the Q95V contacts of each cell, we compute shape features: height, sphericity, and planarity. The comparison shows that the COS-7 OMM–ER linker–induced contacts have a markedly different shape signature compared to those present in HT-1080 and Gp78 overexpressing COS-7 or HeLa cells (i.e., riboMERCS). Averaged over cell, two-sided non-parametric Mann–Whitney test, $n$ = 3 independent biological replicates, ≥30 cells/condition per experiment; *P < 0.05; **P < 0.01; ***P < 0.001.

Shape feature analysis of the largest 5% MERCs shows that riboMERCs of HT-1080 cells are taller, more spherical, and less planar than the MERCs induced by the OMM–ER linker in COS-7 cells (Fig. 8 B). Features of large MERCs induced by

overexpression of Gp78 in COS-7 or HeLa cells are similar to those of HT-1080 riboMERCs, demonstrating that Gp78 induces the formation of larger, more spherical riboMERCs in COS-7 and HeLa cells (Fig. 8 B). Shape features of the larger riboMERCs

induced by the Gp78 Ring finger mutant in HeLa cells were similar to control, indicating the importance of Gp78 ubiquitin ligase activity for the formation of large riboMERCs (Fig. S4 C).

RiboMERCs of HT-1080 cells and those induced by Gp78 expression in COS-7 and HeLa cells form extended, tubular structures that are intercalated between and form contacts with multiple mitochondria (Fig. 9; and Videos 1, 2, 3, 4, 5, 6, 7, 8, 9, 10, 11, and 12). The large MERCs induced by the OMM–ER linker form more elongated planar structures that are closely associated with an individual mitochondrion, similar to the cap-like structures observed by EM upon expression of this linker (Csordás et al., 2006). MCS-DETECT is therefore able to distinguish MERCs based on size and shape features defining the riboMERCs of HT-1080 cells as extended, tubular matrix-like structures that interact with and intercalate between multiple mitochondria. Detection of riboMERC structural changes by MCS-DETECT highlights the importance of whole-cell 3D analysis of MERCs, as these changes could not be detected in our 2D EM analysis, illustrating the potential for discovery MCS-DETECT brings to the study of MERCs.

## Discussion

MERCs were initially characterized based on biochemical identification of mitochondria-associated membranes that identified a role for MERCs in lipid synthesis (Vance, 1990, 1991). MERCs were subsequently shown to play critical roles in ER–mitochondria calcium transport, cellular calcium homeostasis, and apoptosis (Csordás et al., 2018; Herrera-Cruz and Simmen, 2017; Rowland and Voeltz, 2012). However, while the use of functional reporters and cell fractionation approaches to define ER–mitochondria contacts led to an important understanding of the role and composition of MERCs, these approaches are unable to localize MERCs within the cell (Scorrano et al., 2019). Reliance on EM to detect MERCs and associated difficulties in localizing tethers to MERCs has led to discordant results as to whether specific proteins, for instance MFN2, are indeed MERC tethers (Cosson et al., 2012; de Brito and Scorrano, 2008; Dentoni et al., 2022; Leal et al., 2016; Naon et al., 2016; Wang et al., 2015).

Moving beyond EM to use fluorescent microscopy to characterize MERC composition and dynamics includes the development of split fluorescent reporters (such as split-GFP-based contact site sensors; Calì and Brini, 2021; Cieri et al., 2018; Vallese et al., 2020) whose expression may however alter MERC size and stability. The use of fluorescent colocalization to detect MERCs is particularly problematic as the distance between the ER and mitochondria at MERCs (10–80 nm) is far below the diffraction limit (200–250 nm) of visible light used for fluorescent microscopy. Further, the poorer axial resolution of most fluorescent microscopy approaches, including super-resolution, has restricted fluorescent microscopy analysis of MERCs to peripheral ER tubules (Friedman et al., 2011) such that analysis of mitochondria interaction within the convoluted perinuclear ER remains poorly understood. Here, using STED super-resolution imaging and a novel segmentation-free subpixel resolution approach to identify MCS (MCS-DETECT), we identify MERCs in whole-cell 3D volumes of HT-1080, COS-7, and HeLa cells and

further detail the role of Gp78 ubiquitin ligase activity and the riboMERC tether RRBP1 in the abundance and morphology of riboMERCs.

Current image segmentation using fixed thresholds faces challenges of varying background density, difficulty of thresholding signals of varying density, and is highly subjective and user dependent. In addition, analysis of overlap between two channels is based on separate segmentation of independent channels, such that error in capturing interaction, in this case MERCs, is unknown. An optimal method adapts automatically to the data, ensuring consistent results, across images and datasets. To this end, we developed a multichannel self-tuning object detection method (SPECHT) building upon our work in SMLM object detection (Efficient Recurrent Graph Optimized Emitter Density Estimation in Single Molecule Localization Microscopy; Cardoen et al., 2020, 2022). To enhance object (spot) detection, SPECHT scans the image and applies a Laplacian differential operator to detect local signal differentials and identify object edges. Automated image scanning (a sliding window) and kurtosis-based image alignment result in highly sensitive detection, independent of variations in local signal or background intensity differentials. SPECHT has been applied to monitor autophagic flux using tandem fluorescent LC3 (Alan et al., 2022) and is optimal to segment the ER, an organelle of highly varied density and complexity. To validate our object detection method, we generate complex in silico datasets to accurately detect interaction of objects in low SNR channels, especially where the SNR varies markedly between channels. To identify interaction zones between two channels, we identify overlapping negative intensity differentials of both channels (mitochondria, ER); overlap is only detected where edges of the two signals are in sub-resolution space (i.e., lower than the 3D STED resolution of ~100 nm). Laplacian operators are dependent on local signal differential for each channel such that detection of overlapping regions is independent of signal intensity. The Spearman correlation of overlapping ER and mitochondrial Laplacians defines MERC regions that are described by volume and voxel-wise Spearman response. MCS-DETECT therefore represents a novel subpixel resolution approach ideally suited to detect contact zones between organelles.

Morphologically distinct riboMERCs are wider, include ribosomes in the intervening space, and are abundantly expressed in liver (Anastasia et al., 2021; Csordás et al., 2006). They are less abundant in most cultured cells, as shown here in our EM analysis of COS-7 cells, but were found to be abundant by EM analysis of HT-1080 fibrosarcoma cells (Wang et al., 2015) that express robust amounts of the Gp78 ubiquitin ligase (Alan et al., 2022; Tsai et al., 2007). Gp78 knockdown in HT-1080 cells selectively reduced riboMERC expression (Wang et al., 2015), a result we confirm here using CRISPR/Cas Gp78 KO HT-1080 cells. The demonstration here that WT Gp78 but not a Ring finger mutant Gp78 can induce large riboMERCs in COS-7 and HeLa cells defines a role for Gp78 ubiquitin ligase activity in riboMERC formation. Gp78 regulation of riboMERCs may involve interaction with ubiquitinated mitochondrial partners, such as the mitofusins (Fu et al., 2013; Mukherjee and Chakrabarti, 2016). However, the presence of riboMERCs in Gp78 KO cells

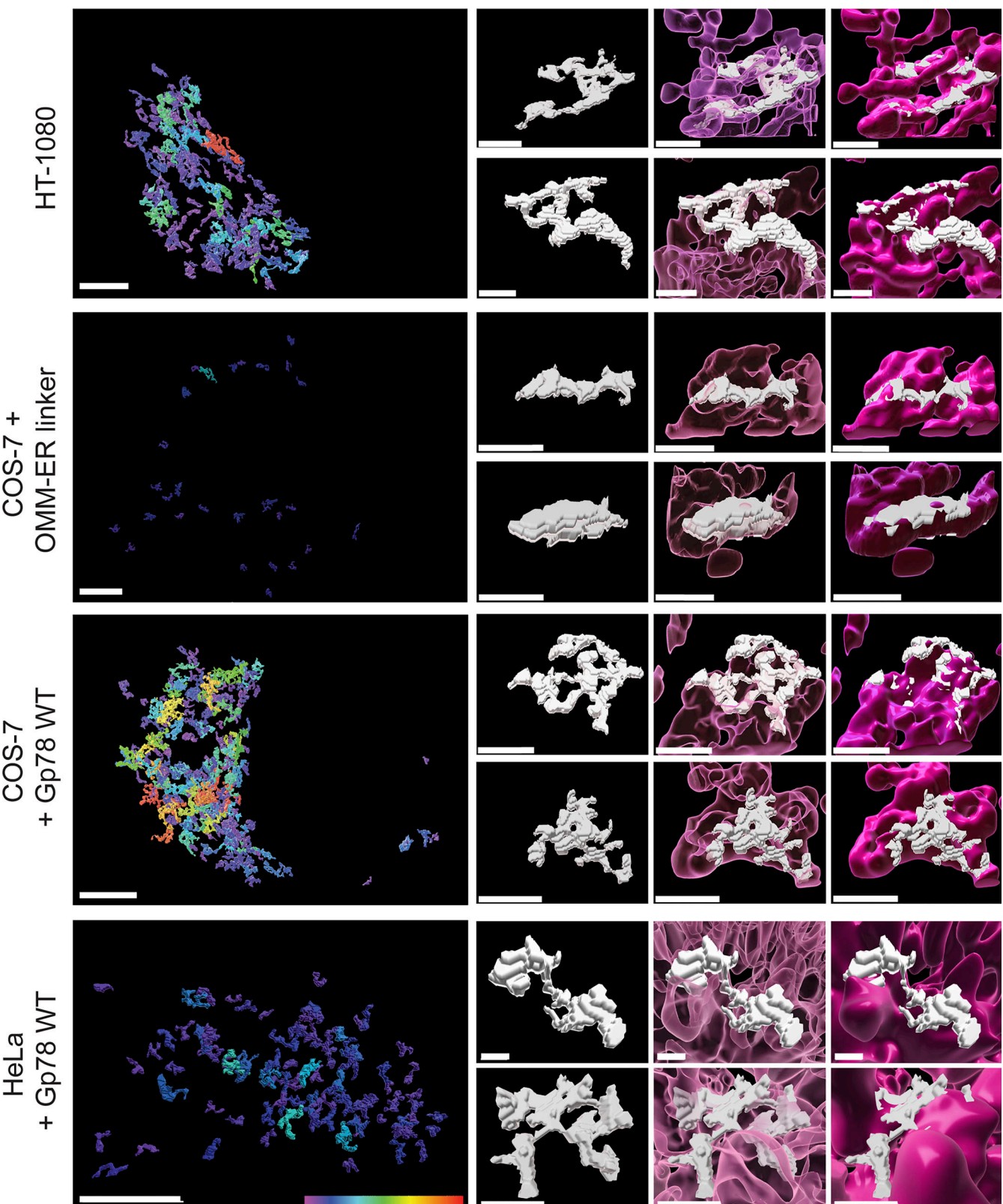

Figure 9. **Gp78 induces convoluted, tubular riboMERCs.** Representative whole-cell views of Q95V MERCs (color-coded for increasing size from 500 to 5,613 voxels) from HT-1080 cells, COS-7, or HeLa cells overexpressing Gp78 WT and COS-7 cells overexpressing the OMM–ER linker as well as representative individual Q95V MERCs alone or adjacent to transparent (pink) or solid mitochondria (magenta) to highlight intercalation of riboMERCs with mitochondria. Rotating videos of these MERCs are included as Supplemental Videos 1, 2, 3, 4, 5, 6, 7, 8, 9, 10, 11, and 12. Bar = 1 µm.

clearly shows that other factors promote the formation of riboMERCs independently of Gp78. Interaction between the OMM protein SYNJ2BP/OMP25 and its ER partner, RRBP1, was shown to induce the formation of riboMERCs in HEK293 cells (Hung et al., 2017). Knockdown of RRBP1 results in reduced riboMERC formation and increased spacing of riboMERCs in liver (Anastasia et al., 2021). Our demonstration that RRBP1 knockdown in HT-1080 eliminates riboMERCs defines an essential role for RRBP1 in the formation of riboMERCs. RRBP1 tethering activity, through interaction with its mitochondrial partner SYNJ2BP (Hung et al., 2017), occurs independently of Gp78. Regulation of RRBP1-dependent riboMERCs by Gp78 ubiquitin ligase activity therefore represents a novel paradigm for MERC formation, in which an effector protein modulates the extent to which a MERC tether can form contact sites.

3D EM tomography studies of riboMERCs in liver show that they wrap completely around mitochondria, covering 30–100% of associated mitochondria, and were therefore named WrappER (Anastasia et al., 2021). In contrast, riboMERCs in HT-1080 cells do not wrap completely around mitochondria. By thin section EM (Fig. 1), riboMERCs form extended contacts along a single face of the mitochondrion that cover at most 30% of the mitochondrion perimeter (Fig. 1). While WrappER in liver present a sheet-like morphology (Anastasia et al., 2021), 3D STED imaging shows that Gp78-dependent riboMERCs form convoluted, tubular structures resembling tubular matrices (Nixon-Abell et al., 2016) that intercalate between and interact with multiple mitochondria. Consistent with a tubular morphology, the MERCs of HT-1080 and Gp78-transfected COS-7 and HeLa cells show increased contact sites per analysis window relative to control cells. In contrast to the tubular Gp78-dependent riboMERCs, MERCs induced by the OMM–ER linker formed more planar structures and presented a decreased contact number relative to riboMERCs; this is consistent with the tight (<20 nm) extended planar contacts over the surface of a mitochondrion formed upon expression of this interorganellar linker (Csordás et al., 2006). An increased number of contacts per sliding window could reflect the varied distance between ER and mitochondria within the contact, as observed for liver riboMERCs upon knockdown of the riboMERC tether RRBP1 (Anastasia et al., 2021; Hung et al., 2017). By EM, the contact width of HT-1080 riboMERCs is very constant, ranging from 50 to 60 nm along the length of the contact (Wang et al., 2015). We interpret this to mean that HT-1080 riboMERCs form extended tubular networks that form multiple contacts with mitochondria.

As for tubular matrix ER and ER sheets (Nixon-Abell et al., 2016; Sun et al., 2020), the functional and morphological relationship between tubular riboMERCs and sheet-like WrappER remains to be determined. WrappER is implicated in lipid homeostasis and recently shown to interact with peroxisomes in liver (Anastasia et al., 2021; Ilacqua et al., 2022), while Gp78 ubiquitin ligase activity in HT-1080 cells mediates basal mitophagy, promoting mitochondrial health and reducing mitochondrial ROS (Alan et al., 2022). Using mitochondrial potential as a functional output, we confirm here that Gp78 ubiquitin ligase activity increases mitochondrial potential, in contrast to the lack of effect on mitochondrial potential by the OMM–ER linker. Further

confirmation of a role for riboMERCs in promoting mitochondrial potential comes from reduced mitochondrial potential upon knockdown of the riboMERC tether RRBP1 in both WT and Gp78 KO cells, paralleling its effect on riboMERC expression in these cells. Further definition of these and, potentially, other functions for riboMERCs requires further study. Indeed, this study highlights the fact that MERCs can take on diverse morphologies. While this analysis focused on a select subset of riboMERCs and the riboMERC regulators Gp78 and RRBP1, the large number of MERC tethers identified suggests a high degree of diversity of MERCs, with respect to both function and structure (Giacomello and Pellegrini, 2016; Herrera-Cruz and Simmen, 2017).

Study of the diversity of MERCs, their associated tethers, and specific functionality has been limited by the absence of a robust fluorescence-based approach to detect MERCs. Application of MCS-DETECT to 3D super-resolved volumes of fluorescently labeled cells now provides a means to extend our characterization of MERC diversity, localize MERC tethers to contact sites, and study MERC dynamics, complementing other criteria used to define membrane contact sites (Scorrano et al., 2019). Importantly, MCS-DETECT is not dependent on expression of MERC reporter molecules and, relative to EM tomography or FIB-SEM, is able to rapidly analyze multiple cell volumes. While we used here 3D STED fixed cell analysis of established models for riboMERC expression to validate the approach, MCS-DETECT reports on negative Laplacians of any two overlapping fluorescent signals and can easily be applied to contacts between any two organelles detected with any voxel-based super-resolution system.

## Materials and methods

### Cell culture

HT-1080 (Cat# 300216/p517_HT-1080, RRID:CVCL_0317; CLS) and HeLa cell lines were acquired from ATCC and authenticated by short tandem repeat profiling at the TCAG Genetic Analysis Facility (Hospital for Sick Kids, Toronto, ON, Canada https://www.tcag.ca/facilities/geneticAnalysis.html). COS-7 cell line (Cat# 605470/p532_COS-7, RRID:CVCL_0224; CLS) was acquired from ATCC and gifted from Ann-Marie Craig (University of British Columbia, Vancouver, Canada). All cell lines were tested regularly for mycoplasma infection by PCR (ABM). COS-7, HeLa, HT-1080, and Gp78 KO HT-1080 (Alan et al., 2022) cells were grown at 37°C with 5% $CO_2$ in complete RPMI 1640 (HT1080 WT and Gp78 KO) or DMEM (COS7 and HeLa; Thermo Fisher Scientific) containing 10% FBS (Thermo Fisher Scientific) and 1% L-glutamine (Thermo Fisher Scientific) unless otherwise stated. For Mitoview633 experiments, COS-7 and HeLa cells were cloned by limiting dilution to obtain cell populations expressing consistent low mitochondrial potential. Plasmids were transfected using Effectene (Qiagen) according to the manufacturer's protocols for 24 h. siRNA was transfected using Lipofectamine 2000 (Cat#: 11668019; Thermo Fisher Scientific) according to the manufacturer's recommendation for 48 h.

### Antibodies, plasmids, and chemicals

ERmoxGFP plasmid was a gift from Erik Snapp (plasmid # 68072; Addgene), and the OMM–ER linker plasmid (mAKAP1(34–63)-

mRFP-yUBC6) was a gift from György Hajnóczky (Thomas Jefferson University, Philadelphia, PA, USA) (Csordás et al., 2006). EGFP was purchased from Clontech Laboratories, Inc. (#637402). Gp78 WT (pcDNA-Flag-Gp78) and RM (pcDNA-Flag-Gp78 with C536S mutation) plasmids (St-Pierre et al., 2012) and Gp78-IRES-GFP (Flag-Gp78 cloned into pIRES-GFP) and Gp78 Ring finger mutant-IRES-GFP (Flag-Gp78 C536S cloned into pIRES-GFP; Fu et al., 2013) were as described. Antibodies were as follows: mouse anti-TOM20/TOMM20 (ab56783; Abcam); rabbit anti-Flag (F7425; Sigma-Aldrich); mouse anti-β-actin (A5441; Sigma-Aldrich); rabbit anti-RRBP1 (ab95983; Abcam); Alexa-Fluor 568 conjugated goat anti-rabbit (A11036; Molecular Probes); Alexa-Fluor 532 conjugated goat anti-mouse (A11002; Molecular Probes); and goat anti-rabbit-HRP (111-035-003; Jackson Immunoresearch). Goat serum (Cat#: 16210-064) was purchased from Thermo Fisher Scientific. 16% paraformaldehyde (Cat#: 15710) and 25% glutaraldehyde (Cat#: 16220) were from Electron Microscopy Sciences. RRBP1-targeted (5′-CGA UGAAGUAAACGCCUUA-3′, 5′-CAUGAUAGGAGGAAACGAA, 5′-CAGAAUAUAUGGAAGACGU-3′, 5′-GGAAGUUCUUAGUGC UAGA-3′) and scrambled control siRNA (5′-CGAUGAAGUAAA CGCCUUA-3′, 5′-CAUGAUAGGAGGAAACGAA-3′, 5′-CAGAAU AUAUGGAAGACGU-3′, 5′-GGAAGUUCUUAGUGCUAGA-3′) were purchased from Dharmacon (Cat#: L-011891-02-0010). MitoView 633 (#70055) was purchased from Biotium. Other chemicals were from Sigma-Aldrich.

## EM
Cells grown on ACLAR film (Ted Pella) for 24 h until 80–90% confluency, where indicated transfected with specific siRNAs for an additional 48 h, were (1) washed with PBS at room temperature, (2) fixed with 1.5% paraformaldehyde and 1.5% glutaraldehyde in 0.1 M sodium cacodylate (Electron Microscopy Sciences), pH 7.3, for 1 h at room temperature followed by three washes with 0.1 M sodium cacodylate, pH 7.3, (3) postfixed with 1% osmium tetroxide in 0.1 M sodium cacodylate buffer for 1 h at 4°C on ice following by three washes with $H_2O$, (4) stained en bloc with aqueous 1.0% uranyl acetate for 1 h on ice, (5) progressively dehydrated through an ethanol series (50%, 70%, 90%, and 100%) and 100% propylene oxide (Electron Microscopy Sciences) each for 10 min, (6) infiltrated with 1:1 mixture of EMBED 812 (Electron Microscopy Sciences) and propylene oxide, and embedded in EMBED 812, (7) polymerized for 2 d, (8) sectioned into ~900-A-thick slices using Leica Ultramicrotome (Leica Microsystems), (9) stained with uranyl acetate and lead citrate, (10) imaged either on a Tecnai G2 (FEI) or a Talos L120C (ThermoFisher) transmission electron microscope (acceleration voltage: 120 kV), using an Eagle 4k CCD camera (FEI) or a CETA camera (Thermo Fisher Scientific), respectively. Quantification of EM images was done with Image Pro 3Ds based on manual annotation of ER and mitochondria using an Intuos.5 tablet and stylus (Wacom). Mitochondrial membranes were identified due to the presence of cristae. MERCs were identified as regions where the ER membrane came within 80 nm of the mitochondria, the maximum MERC width reported (Giacomello and Pellegrini, 2016). MERC length was assessed as the distance along which the apposition distance between the ER and mitochondria remained constant. riboMERCs were defined based on the presence of at least one ribosome in the inter-organellar space of the contact site.

## Immunofluorescence labeling
Cells grown on #1.5H coverslips (Paul Marienfeld) were (1) fixed with 3% paraformaldehyde with 0.2% glutaraldehyde at room temperature for 15 min and washed with PBS-CM (phosphate buffered saline [PBS] solution supplemented with 1 mM $CaCl_2$ and 10 mM $MgCl_2$; two quick washes and then two 5-min washes); (2) permeabilized with 0.2% Triton X-100 for 5 min then washed with PBS-CM as above; (3) quenched with 1 mg/ml of $NaBH_4$ for 10 min and washed with PBS-CM; (4) blocked with 10% Goat Serum (Thermo Fisher Scientific) and 1% BSA (Sigma-Aldrich) in PBS-CM for 1 h; (5) incubated with primary antibodies in antibody buffer (1% BSA, 2% goat serum, 0.05% Triton-X100, 20X sodium/sodium citrate buffer in Milli-Q $H_2O$) overnight at 4°C, then washed quickly with PBS-CM three times for 5 min with antibody wash buffer (20× saline-sodium citrate solution, 0.05% Triton-X100 in Milli-Q $H_2O$); (6) incubated with secondary antibodies in antibody buffer for 1 h and then washed quickly with PBS-CM six times for 10 min with antibody wash buffer on a rocker; (7) rinsed with Milli-Q $H_2O$ and mounted with ProLong Diamond (Thermo Fisher Scientific) and cured for 24–48 h at room temperature. To assess plasmid expression levels in COS-7 and HeLa, cells were passed, transfected, and labeled in parallel and confocal images acquired with equivalent acquisition settings using a Leica TCS SP8 X White Light Laser Confocal with a 100×/1.4 Oil HC PL APO CS2 objective, white light laser, HyD detectors, and Leica Application Suite X (LAS X) software. GFP was excited at 488 nm while Alexa Fluor 532 was excited at 528 nm and Alexa Fluor 568 at 577 nm. The fluorescent intensity of Flag-Gp78 WT and RM (anti-Flag, Alexa568) and the OMM–ER linker (RFP) was determined using ImageJ (RRID:SCR_003070). Regions of interest (ROIs) for each cell were drawn manually based on segmentation using the maximum brightness. Later, Otsu thresholding was applied to each individual cell and integrated densities of all objects were quantified and normalized by the total area of the FLAG-568 or RFP label per cell (RRID:SCR_003070).

## Gated stimulated emission depletion (gSTED) microscopy
gSTED imaging was performed at room temperature with the 100×/1.4 Oil HC PL APO CS2 objective of a Leica TCS SP8 3X STED microscope (Leica) using white light laser excitation, HyD detectors, and Leica Application Suite X (LAS X) software. Time-gated fluorescence detection was used for STED to further improve lateral resolution. For double-labeled fixed samples, the acquisition was done at a scan speed of 600 Hz with a line average of 3. GFP was excited at 488 nm and depleted using the 592 nm depletion laser. Alexa Fluor 568 was excited at 577 nm and depleted using the 660 nm depletion laser. Sequential acquisition (in the order of AF568/GFP) of stacks at a step size of 100 nm was used to avoid crosstalk. STED images were deconvolved using Huygens Professional software, Version 21.04 with algorithm Cmle: maximum iterations: 30, SNR: 5; quality threshold: 0.0001 (RRID:SCR_014237; Scientific Volume Imaging).

## Mitochondrial potential

Cells were plated in an ibidi chamber and transfected after 24 h with the indicated plasmids for 24 h or siRNA for 48 h and then labeled with the potential-dependent mitochondrial fluorescent reporter MitoView 633 at a concentration of 50 nM for 30 min, washed three times with warm PBS, and incubated in Molecular Probes Live Cell Imaging Solution. Live-cell imaging was performed at 37°C with a Leica TCS SP8 confocal microscope with a 100×/1.40 Oil HC PL APO CS2 objective (Leica) equipped with a white light laser, HyD detectors, an environmental chamber, and Leica Application Suite X (LAS X) software. Images were analyzed using ImageJ software. ROIs per cell were segmented using the maximum brightness. Later, Otsu thresholding was applied to each individual cell and integrated densities of all mitochondrion objects over 5 μm² were quantified and normalized by the total area of the mitochondrial label, per cell. One-way ANOVA and Tukey post-hoc statistical analyses were applied to the data.

## Western blots

Confluent cells were scraped off the dish in PBS. The cell pellet was lysed using M2 (20 mM Tris HCl [pH 7.6]; 0.5% NP-40; 250 mM NaCl; 3 mM EGTA [pH 8.0], 3 mM EDTA [pH 8.0]) with added protease and phosphatase cocktail inhibitor tablets (Roche) and incubated for 40 min at 4°C. Lysates were centrifuged for 15 min at 13,200 RPM and 4°C to remove cell debris. Protein concentration was determined using a Bradford Assay and equal amounts of protein were loaded onto the sodium dodecyl sulfate-polyacrylamide gel electrophoresis gel and run at 135 V and 2.00 A for ~1 h. The proteins were transferred to a methanol-activated polyvinylidene fluoride membrane using a semi-dry BioRad transfer system. The membrane was subsequently placed in PBS-T (PBS supplemented with 0.1% Tween-20) with 10% glutaraldehyde on a shaker for 30 min at room temperature. The membrane was blocked with 5% milk product in PBS for 1 h and then incubated with appropriate concentrations of the respective primary antibodies overnight at 4°C. The following day the membranes were washed and incubated with either rabbit or mouse HRP-conjugated secondary antibodies for 1 h at room temperature. The membrane was washed three times with PBS-T for 10 min each and blots developed with enhanced chemiluminescence and protein bands visualized using film or Chemidoc (BioRad). Where indicated, images were subjected to densitometric quantification and statistical analysis (one-way ordinary ANOVA) using GraphPad (Prism 9).

## MCS-DETECT: Contact zone detection pipeline

The source code is available under an open-source license (AGPLv3) at https://github.com/bencardoen/SubPrecisionContactDetection.jl. An optimized version with all dependencies in a single container image is provided as well for ease of use.

Illustrated in Fig. 2 C, the algorithm is composed of three steps: computing correlation of intensity and Laplacian for both channels, computing the confidence map, and finally filtering to remove artifacts. A full listing in pseudocode of each stage is provided in Fig. S1 B.

## Spearman correlation

We retain the negative Spearman response on the negative Laplacian of each channel (Fig. S1 B, Algorithm 1, 3). To prevent the inclusion of low-intensity signals colocalizing with high-intensity signals, we compute the negative Spearman response of the intensity of both channels; only where both Laplacian and intensity correlate, do we retain the Laplacian correlation.

## Confidence map

For each Spearman response voxel, we compute its significance and the minimum observable correlation (Fig. S1 B, Algorithm 4, 5), given the sample size (window), using a z-test. Typical values used here are 0.05 both for alpha and statistical power (1-beta). In Fig. S1 A, we show that this parameter leads to consistent results on representative cells.

## Filtering

Because 3D STED microscopy is often anisotropic in its point spread function, and one thus risks "bleed through" across z-planes, we apply a bleed-through filter (Fig. S1 B, Algorithm 2) as a mask by filtering out intensity lower than a given z-score (standard score of intensity), where z is the parameter set in the filter. Bleed-through intensity values will have markedly smaller z-values compared with ER and mitochondria segments. Because the z-score is a pivotal quantity, it will adapt to each channel's distribution and thus lead to an unbiased filter. We test the effect of this filter in Fig. S1 B, where consistent results are retained across a large range of parameters. Due to low SNR and pixelation, a "shadowing" response, parallel at an offset of three to four voxels to a true response, can appear. As can be observed from the Laplacian curves in Fig. 2 B (middle row), voxels where only Laplacian of only one channel changes cannot colocalize with a contact. Our gradient filter computes the change of the Laplacian by way of the third derivative of intensity. Next, we mask out any voxels where this third derivative is zero for one channel, yet nonzero for the other. The interested reader can infer from Fig. 2 B that a third derivative can be zero for a valid contact, but only if both are zero.

To ensure that we report on contacts with whole mitochondria and not mitochondria fragments or vesicles, as we do by EM, we exclude contacts that are adjacent to mitochondria below a size and mean intensity threshold of, respectively, 9 (ln scale, voxels) and 0.2, chosen based on empirical observation of non-target mitochondria-labeled segments. We illustrate that these thresholds split the distribution of mitochondria between mitochondria and vesicle-like objects in Fig. S2. For COS-7 and HT1080 cells, a clear bimodal distribution of mitochondria segments is separable. 3D surface projections of filtered image stacks were generated using BitPlane Imaris V10.

## Quantifying contacts

We compute features of contacts sampled by both a sliding window and per-contact. Contacts are distributed with a frequency strongly inverse to their size. Quantifying such long tail distributions can undercount the visually more apparent large contacts. To this end, we slide a non-overlapping window over the contact channel and record size and frequency per cube,

estimating local density. At the whole contact level, we record the confidence, shape, size, and spatial distribution of the contacts. The resulting two sets of features allow the end user a local and global descriptive view of contacts in the cell.

For our data, we use a sliding window of 25 × 25 × 5 voxels, which equates to a cube given the anisotropy of the acquisition. We show in Fig. S2 A how this window size does not alter the consistency of our results. We average results over cells and test the result with a non-parametric two-sided Mann–Whitney test. Applying multiple testing corrections is not practical here, given that we have pairwise (HT-1080) and three pairwise comparisons (COS-7). Were the correction to be applied only to COS-7, we would be reporting results of COS-7 with a risk of higher false negatives than for HT-1080. Contacts are analyzed both by sampling the image and also as a whole. Large contacts are defined as those exceeding the 95th quantile of volume, per cell. Contacts with size ≤2 voxels are removed, as they are below the diffraction limit. We compute shape features (sphericity, anisotropy, and planarity) based on the eigenvalues of the Spearman-weighted contacts. Not reported but provided to users of MCS-DETECT are confidence of reconstruction, clustering, and position in Z, to name a few.

### Statistical analysis

Statistical tests on the output of the contact detection pipeline are always computed on aggregated (per cell) results to avoid breaking the independence assumption or inducing Simpson's paradox. No outlier removal is performed. Plots are edited for cosmetic alterations only. To avoid dependence on the (partial) presence of the normal distribution, we use the non-parametric two-sided Mann–Whitney test for comparison. 2D kernel density estimation (KDE) is used to gain insight into potentially interesting patterns in 2D distributions in addition to the regression lines. Per replicate estimation, due to the cost of acquiring 3D STED data, would be too low powered (≥10 cells per replicate, minimum 3 replicates per condition) to quantify separately. To capture robustly the within-cell variance of long-tail distributed contacts, we quantify both mean/cell and the 95th quantile, ensuring long-tail behavior does not induce significance where not justified.

### Runtime and memory constraints and its consequences for parameters

The detection algorithm uses a window of size w for 3D, leading to a total window size of $(2*w+1)^3$ voxels. For our results, we set w to 2, so each window is 125 voxels large. Let N = X × Y × Z, the dimensions of both channels, and M = $(2*w+1)^3$. Time complexity is dominated by the Spearman correlation, which needs to be applied N times, leading to $O(N\ M \log M)$ complexity. Space complexity is linear in N; however, to offer the end user maximal interpretability, we output multiple intermediate stages, leading to an estimated memory use of ∼10 * N. For our data, depending on the cell size, memory usage averages between 32 and 128 GB. The complexity analysis is essential to illustrate why we set w = 2. A window size determines directly the minimum observable Spearman response. Larger windows therefore can detect fainter responses. However, a large window risks including multiple interaction patterns, for example, when an interaction zone is enclosed by two mitochondria. In such cases, the response of both would cancel out the desired response, and we would lose information. The complexity analysis shows that increasing w leads to a cubic increase in runtime. All of these reasons lead us to use a window of 2, spanning ∼350 nm.

### Limitations

It is important to frame the proposed contribution within its limitations. First, any misaligned or unregistered channels can induce false responses or, more likely, cause contact detection to fail. Normalization of both input channels can destroy reconstruction and is not needed, as each stage is already adapting to the relative intensity distribution. Similarly, if the deconvolution is not calibrated correctly to match the empirical PSF, reconstruction can be compromised. Second, the recovery of sub-precision information is only possible because we exploit the local differential intensity profile of each voxel over a given window. If the SNR is too low, differential analysis can break down. If the precision of the system decreases, or spans many voxels, the localization of the interface will become more challenging. For example, from Fig. 2 B it can be deduced that if the two intensity profiles widen too much, the negative correlation vanishes when their negative second differentials no longer overlap. This phenomenon will become more likely when object size decreases with respect to the width of the interface. For instance, while MCS-DETECT can in principle be applied to diffraction-limited samples, recall of interfaces would be less precise. As shown in Figs. 3, 4, and 5, twofold improved resolution of 3D STED image stacks effectively enabled detection of MERCs, which present a 3- to 10-fold reduced width compared to the resolution of the system. Similarly, a too-large window size can destroy a response, for instance when the window spans the size of both objects. In this case, positive and negative differentials cancel each other out. Finally, anisotropic precision leads to responses that are less precise in the axis where precision is lowest. In our case precision is worst in Z, so recovered contact sites on top of mitochondria are likely to be captured with reduced precision. Finally, note that MCS-DETECT does not separate proximate contacts that a user could segment as two or more individual adjacent contacts. In future work, a more refined approach will tackle the per-region identification even for proximate contacts, but this is limited by ground truth voxels annotated for specific contacts.

### Online supplemental material

Fig. S1 shows parameter study of the proposed method. Fig. S2 shows identifying and filtering mitochondria. Fig. S3 shows Gp78 overexpression western blot and representative images of MitoView 633 experiments. Fig. S4 shows Gp78 expression in COS-7 and HeLa cells, siRRBP1 western blot, and shape features for Q95 Gp78 RM overexpressing cells. Videos 1, 2, 3, and 4 show 360° views of MERCs larger than the 500 voxels in complete HT-1080 cell, COS-7 cell overexpressing the OMM–ER linker or Gp78 WT, or HeLa cell overexpressing Gp78 WT. Videos 5, 6, 7, 8, 9, 10, 11, and 12 depict 360° views of individual 95th quantile MERCs in HT-1080 cells, COS-7 cell overexpressing the OMM–ER linker or Gp78 WT, or HeLa cell overexpressing Gp78 WT.

## Data availability

All data are available upon request. The source code is available under AGPLv3 license at https://github.com/bencardoen/SubPrecisionContactDetection.jl.

## Acknowledgments

This study was supported by grants from the Canadian Institutes of Health Research (AWD-022443, PJT-148698, COP-143083, COP-137359) and from the Natural Sciences and Engineering Research Council of Canada (RGPIN-2019-05179, RGPIN-2020-06752, RGPIN-2018-03727). All imaging was performed at the University of British Columbia Life Sciences Institute Imaging Core Facility, RRID:SCR_023783.

Author contributions: Conceptualization: B. Cardoen, K. Vandevoorde, G. Gao, M. Ortiz-Silva, G. Hamarneh, and I.R. Nabi. Data Curation: B. Cardoen, K. Vandevoorde, G. Gao, and M. Ortiz-Silva. Formal Analysis: B. Cardoen, K. Vandevoorde, G. Gao, M. Ortiz-Silva, and P. Alan. Funding Acquisition: G. Hamarneh and I.R. Nabi. Investigation: B. Cardoen, K. Vandevoorde, G. Gao, M. Ortiz-Silva, W. Liu, E. Tiliakou, and A.W. Vogl. Methodology: B. Cardoen, K. Vandevoorde, G. Gao, M. Ortiz-Silva, P. Alan, A.W. Vogl, G. Hamarneh, and I.R. Nabi. Project Administration: G. Hamarneh and I.R. Nabi. Resources: G. Hamarneh and I.R. Nabi. Software: B. Cardoen and G. Hamarneh. Supervision: G. Hamarneh and I.R. Nabi. Validation: B. Cardoen, K. Vandevoorde, G. Gao, M. Ortiz-Silva, P. Alan, A.W. Vogl, G. Hamarneh, and I.R. Nabi. Visualization: B. Cardoen, K. Vandevoorde, G. Gao, M. Ortiz-Silva, W. Liu, A.W. Vogl, G. Hamarneh, and I.R. Nabi. Writing—Original Draft: B. Cardoen, K. Vandevoorde, G. Gao, and I.R. Nabi. Writing—Review & Editing: B. Cardoen, K. Vandevoorde, G. Gao, M. Ortiz-Silva, P. Alan, W. Liu, A.W. Vogl, G. Hamarneh, and I.R. Nabi.

Disclosures: I.R. Nabi reported "Leica has paid travel expenses for seminar presentations at the University of Toronto and Western Ontario, October 2023." No other disclosures were reported.

Submitted: 23 June 2022

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

**Supplemental material**

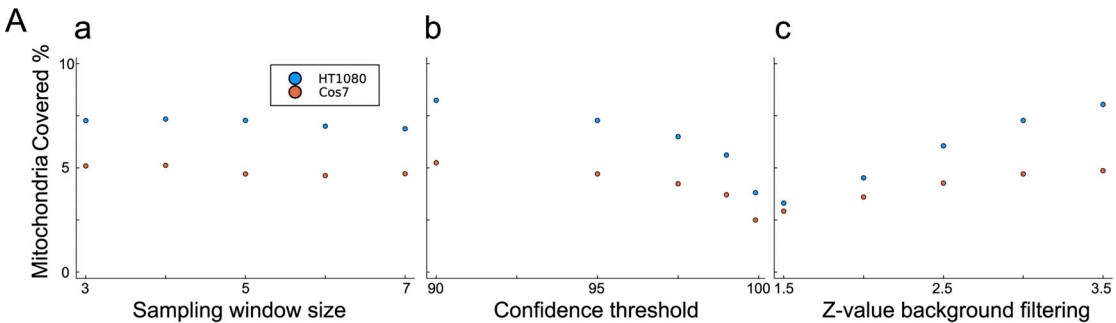

**A**

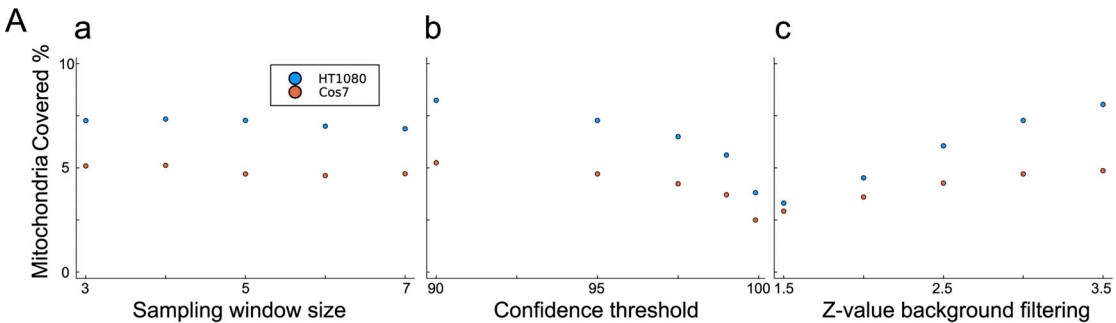

**B**

---

**Algorithm 1** Contact detection algorithm

**Input** $R, G$ : 2x3D, window size $w, \alpha, \beta, \sigma = 1$
**Output** $C_d$: contact prediction, confidence map $M_d$
$R_l, G_l \leftarrow$ Gaussian(Laplacian$(X), \sigma$) for X in $[R, G]$
$C_i, M_i \leftarrow$ spearman$(R, G)$ ▷ Alg.3
$C_d, M_d \leftarrow$ spearman$(R_l, G_l)$
$C_d =$ sigfilter$(C_d, M_d, w, \alpha, \beta)$ ▷ Alg.4
$C_i =$ sigfilter$(C_i, M_i, w, \alpha, \beta)$
$C_d \leftarrow C_d.*(C_d. \wedge C_i)$
$C \leftarrow C_d.*$ gradientfilter$(R_c, G_C)$ ▷ Alg.6
Return $C_d, M_d$

---

**Algorithm 2** Background removal

**Input** $R, G$, background filter strictness z
**Output** $R', G'$
$G'[x, y, z] \leftarrow (G'[x, y, z] > \mu_G + z * \sigma_G) \; ? \; 1 \; : \; 0$
$R'[x, y, z] \leftarrow (R'[x, y, z] > \mu_R + z * \sigma_R) \; ? \; 1 \; : \; 0$
Return $R', G'$

---

**Algorithm 3** Spearman

**Input** $R, G$, window size $w$
**Output** Correlation $C$, Confidence $M$
$N \leftarrow (2 * w + 1)^3$
**for** $x, y, z \in dim(R)$ **do**
 $a, b \leftarrow R[x \pm w, y \pm w, z \pm w], G[x \pm w, y \pm w, z \pm w]$
 $si_a =$ index(sort$(a)$)
 $si_b =$ index(sort$(b)$)
 $r = \frac{cov(si_a, si_b)}{\sigma_{si_a} \sigma_{si_b}}$
 $C[x, y, z] = |$Min$(r, 0)|$
 $M[x, y, z] = \sqrt{\frac{N-3}{1.06}} *$ atanh$(r)$ ▷ z-score of Fisher transform
**end for**
Return $C, M$

---

**Algorithm 4** Significance filter

**Input** $X$, window size $w, \alpha, \beta, M$
**Output** Filtered $X'$
$r_\beta =$ minr$((2 * w + 1)^3, \alpha, \beta)$ ▷ Alg.5
$X'[x, y, z] = X[x, y, z] * (M[x, y, z] < \alpha \wedge X[x, y, z] > r_\beta) \quad \forall x, y, z \in dim(X)$
Return $X'$

---

**Algorithm 5** Minr

**Input** $N$ (sample size), $\alpha, \beta$
Output Minimum observable correlation r
$N' = \infty$
$z_\alpha, z_\beta = qnorm(\alpha, \beta)$▷ Quantile of normal distribution
**for** $r \in [0, 1]$ **do**
 z' $= \frac{1}{2} \log \frac{1+r}{1-r}$
 N' $= \left(\frac{z_\alpha + z_\beta}{z'}\right)^2 + 3$
 **if** $N' < N$ **then**
 **Return** r
 **end if**
**end for**
**Return** 1

---

**Algorithm 6** Gradient filter

**Input** $R, G$
**Output** Mask where gradient detects no artifacts
$R_d, G_d \leftarrow |\nabla_R^3|, |\nabla_G^3|$ ▷ 3rd derivative intensity
**Return** $\neg((R_d. * G_d. = 0). \wedge (R_d. + G_d.! = 0))$

---

**Algorithm 7** Analysis

**Input** $C, X, Y, Z$, minimal observable size $P$
**Output** Contacts, Features
$C' \leftarrow$ morphological closing(C)
$O \leftarrow$ connectedcomponents(C)
$O' \leftarrow \{o \; | \; |o| > P \quad \forall o \in O\}$
**for** $i \in dim(C)/(X, Y, Z)$ **do**
 $cube \leftarrow O[x + iX, y \pm iY, z \pm iZ]$
 result[i] = analyze(cube)
**end for**
**Return** result, $O$

---

Figure S1. **Parameter study of the proposed method. (A)** We test two representative HT-1080 and COS-7 cells, with known distinct contact types. We vary the analysis window, a 3D cube of 5k × 5k × k over the mitochondria and contact channel. (a) The surface coverage ratio stays stable for a range of k. (b) The user can set a significance and statistical power threshold. This increases the minimum significance at which voxels can be detected, as well as the minimum correlation considered to be observable. As expected, when this threshold increases, expected differences between two representative cells decrease, as does the overall number of correlation voxels. At the limit of 100% confidence, no information would be left. To avoid false responses by bleed-through of signal in nearby Z-planes, as well as high background intensity in low SNR conditions, we apply an adaptive threshold in z-space. (c) We observe that at low values, the inclusion of false responses masks any differences (z = 1.5). At high values (3.5) the ER channel was visually degraded, we see that after z = 3 the difference between the two cells is maximal and converged. **(B)** The full reference algorithm pseudocode listing of each stage of the proposed method, enabling reproduction in any implementation. The actual Julia implementation used adds non-algorithm stages to deal with parallelization, optimization, error handling, and recording intermediate stages, which are out of scope for the purposes of this listing. The source code is available under AGPL v3 license at https://github.com/bencardoen/SubPrecisionContactDetection.jl.

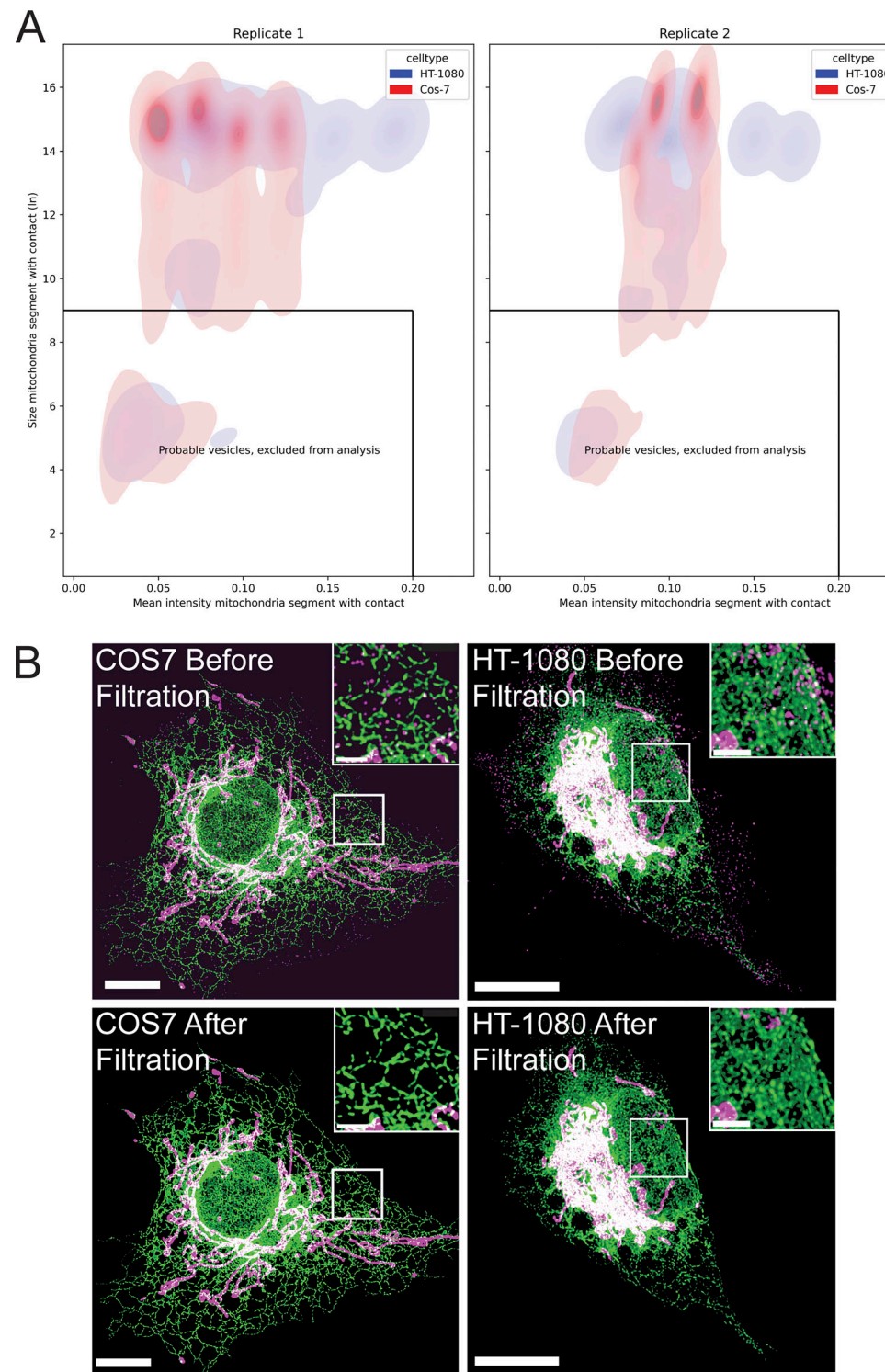

Figure S2. **Identifying and filtering mitochondria. (A)** For each contact, size and mean intensity of the adjacent mitochondria is plotted for two replicates, to indicate consistency across replicates. Note that this is by default larger than a segmentation method would compute. The size and mean intensity of mitochondria in HT-1080 and COS-7 cells present a clearly separable group of small, low-intensity mitochondria structures. To report results on what are clearly and unambiguously mitochondria, corresponding to the mitochondria observed by EM, mitochondrial structures smaller than thresholds 9 (ln size) and 0.2 (mean intensity) were eliminated. **(B)** 3D STED images show labeling of ER (green), mitochondria (magenta), and contact sites (white) of representative COS-7 and HT-1080 cells before and after mitochondrial filtering. Bar = 10 µM (insets: 1 µM).

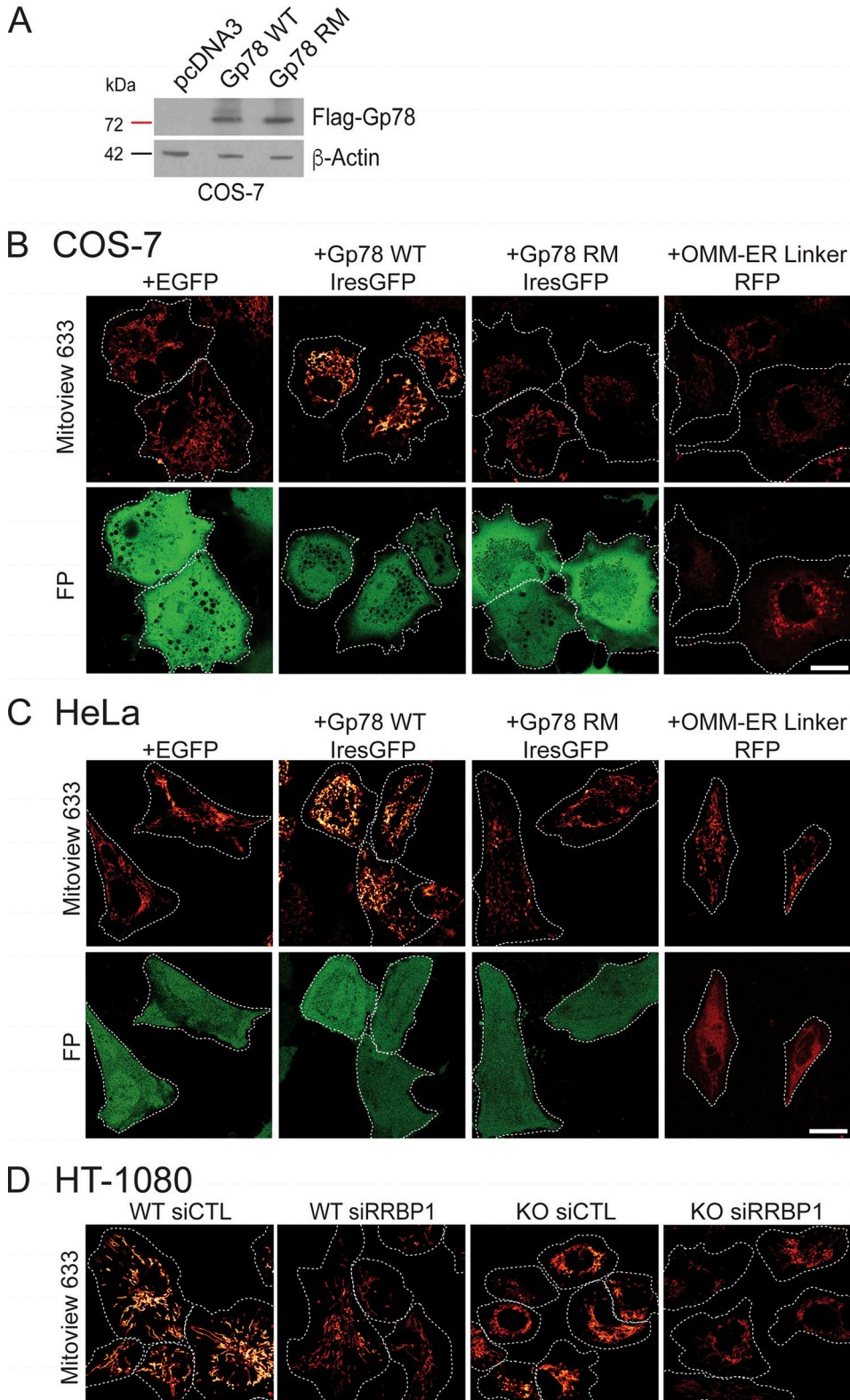

Figure S3. **Gp78 overexpression western blot and representative MitoView images. (A)** Western blots of COS-7 cells transfected with pcDNA3, Flag-Gp78 WT, or Flag-Gp78 RM were probed with antibodies to Flag tag to reveal Gp78 and to β-actin. **(B)** Representative images of COS-7 cells transfected with EGFP (as a control), Gp78 WT IRES-GFP, Gp78 RM IRES-GFP, or the OMM–ER linker (RFP) and labeled with MitoView 633. Corresponding GFP or RFP images (FP) are shown and cell boundaries outlined (Fig. 4 D). **(C)** Representative images of HeLa cells transfected with EGFP (as a control), Gp78 WT IRES-GFP, Gp78 RM IRES-GFP, or the OMM–ER linker (RFP) and labeled with MitoView 633. Corresponding GFP or RFP images (FP) are shown and cell boundaries outlined (Fig. 5 D). **(D)** Representative images of HT-1080 cells transfected with either siControl or siRRBP1 and labeled with MitoView633 are shown and cell boundaries outlined (Fig. 7 C). Bars (B, C, D) = 20 µm. Source data are available for this figure: SourceData FS3.

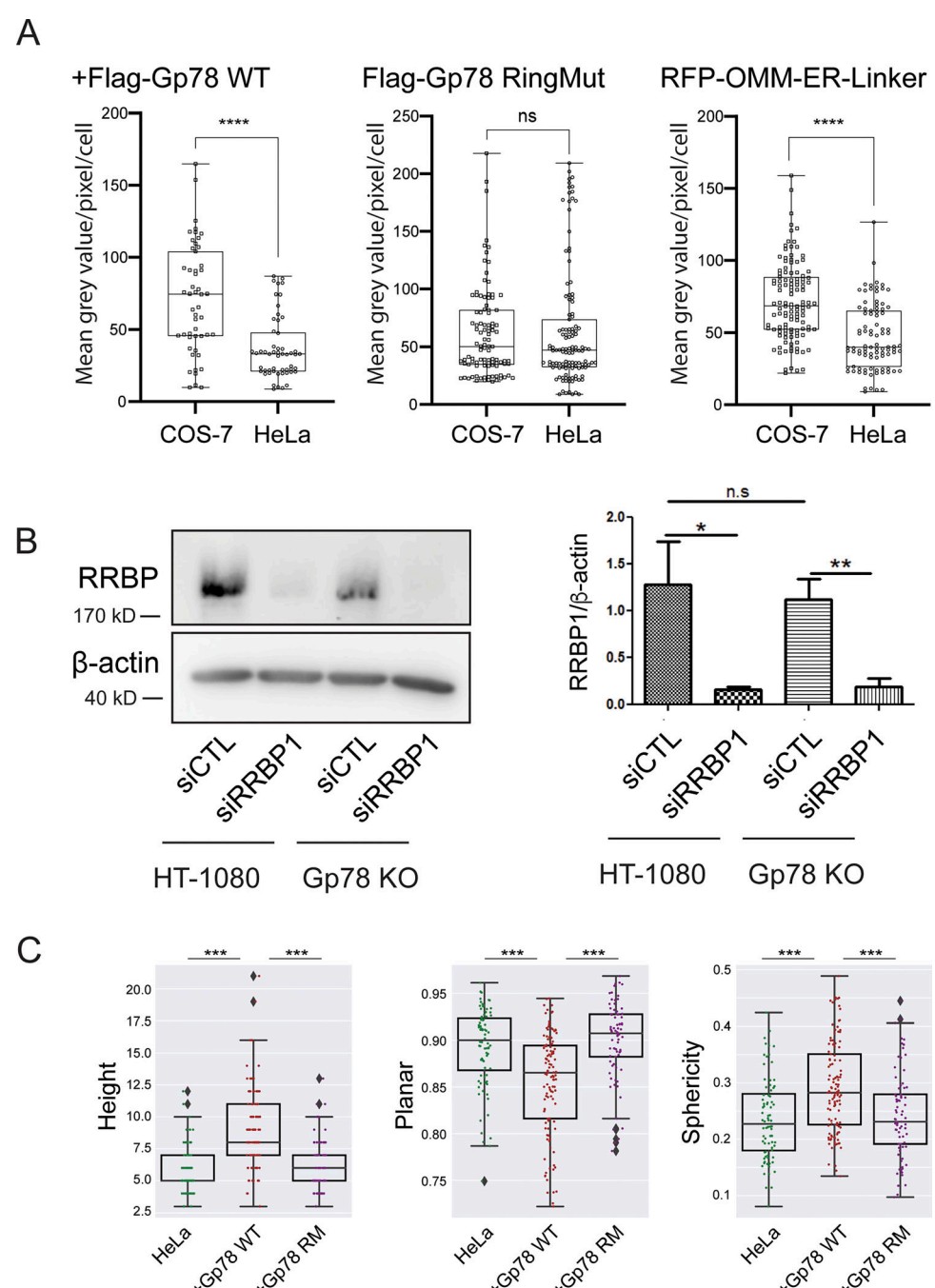

Figure S4.   **Gp78 expression in COS-7 and HeLa cells, siRRBP1 western blot, and shape features for Q95 Gp78 RM overexpressing cells. (A)** COS-7 and HeLa cells were transfected in parallel with Flag-Gp78 WT, Flag-Gp78 RM, or the RFP-tagged OMM–ER linker and fixed after 24 h. The Flag-Gp78 transfected cells were labeled for anti-Flag and the OMM–ER linker transfected cells left unlabeled. Cells were imaged and anti-Flag or RFP labeling density was quantified. $n = 3$; >36 cells per sample; ****$P < 0.0001$; Student $t$ test. **(B)** Western blots of RRBP1 and β-actin show a reduction of RRBP1 in HT-1080 and Gp78 KO HT-1080 cells following transfection of siRRBP1 relative to siCTL. Graph shows densitometric quantification of band intensity. $n = 3$; *$P < 0.05$; **$P < 0.01$. **(C)** Shape features, height, sphericity, and planarity for the Q95V contacts of a representative cell closest to the mean Q95V for control HeLa cells or HeLa cells overexpressing Gp78 WT or Gp78 RM were analyzed. The comparison shows that the large contacts induced by Gp78 RM in HeLa cells have a shape signature comparable to control cells and not to the riboMERCs induced by Gp78 overexpression. Averaged over cell, two-sided non-parametric Mann–Whitney test, $n = 3$; ***$P < 0.001$. Source data are available for this figure: SourceData FS4.

**Video 1. 360° views of MERCs larger than the 500 voxels in complete HT-1080 cell, transfected with ERmoxGFP, labeled for anti-TOM-20, and imaged using 3D STED (**Fig. 9**).** Video was rendered in Imaris 10.0 software using the contacts channel outputted from MCS-DETECT (30 frames/s). MERCs are color-coded for increasing size from 500 (blue) to 5,613 voxels (red).

**Video 2. 360° views of MERCs larger than the 500 voxels in a COS-7 cell overexpressing the OMM–ER linker (**Fig. 9**), transfected with ERmoxGFP, labeled for anti-TOM-20, and imaged using 3D STED (**Fig. 9**).** Video was rendered in Imaris 10.0 software using the contacts channel outputted from MCS-DETECT (50 frames/s). MERCs are color-coded for increasing size from 500 (blue) to 5,613 voxels (red).

**Video 3. 360° views of MERCs larger than the 500 voxels in a COS-7 cell overexpressing WT Gp78-FLAG (**Fig. 9**), transfected with ERmoxGFP, labeled for anti-TOM-20 and anti-FLAG, and imaged using 3D STED (**Fig. 9**).** Video was rendered in Imaris 10.0 software using the contacts channel outputted from MCS-DETECT (30 frames/s). MERCs are color-coded for increasing size from 500 (blue) to 5,613 voxels (red).

**Video 4. 360° views of MERCs larger than the 500 voxels in a HeLa cell overexpressing WT Gp78-FLAG (**Fig. 9**), transfected with ERmoxGFP, labeled for anti-TOM-20 and anti-FLAG, and imaged using 3D STED (**Fig. 9**).** Video was rendered in Imaris 10.0 software using the contacts channel outputted from MCS-DETECT (50 frames/s). MERCs are color-coded for increasing size from 500 (blue) to 5,613 voxels (red).

**Video 5. 360° views of an individual 95th quantile MERC in an HT-1080 cell transfected with ERmoxGFP, labeled for anti-TOM-20, and imaged using 3D STED (**Fig. 9**).** MERCs display a high degree of complexity, with multiple branch points and extending over several Z slices. Video was rendered in Imaris 10.0 software using the contacts, mitochondria, and ER channels outputted from MCS-DETECT (50 frames/s). All channels depict a single contact. Additional channels are rendered after each rotation in the following order: MERC channel (white), transparent mitochondria (pink), opaque mitochondria (red), and transparent ER (green).

**Video 6. 360° views of an individual 95th quantile MERC in an HT-1080 cell transfected with ERmoxGFP, labeled for anti-TOM-20, and imaged using 3D STED (**Fig. 9**).** MERCs display a high degree of complexity, with multiple branch points and extending over several Z slices. Video was rendered in Imaris 10.0 software using the contacts, mitochondria, and ER channels outputted from MCS-DETECT (50 frames/s). All channels depict a single contact. Additional channels are rendered after each rotation in the following order: MERC channel (white), transparent mitochondria (pink), opaque mitochondria (red), and transparent ER (green).

**Video 7. 360° views of an individual 95th quantile MERC in a COS-7 cell overexpressing the OMM–ER linker (**Fig. 9**), transfected with ERmoxGFP and labeled for anti-TOM-20, and imaged using 3D STED.** Planar MERCs follow the mitochondria in a single direction with limited travel to additional Z slices. Video was rendered in Imaris 10.0 software using the contacts, mitochondria, and ER channels outputted from MCS-DETECT (50 frames/s). All channels depict a single contact. Additional channels are rendered after each rotation in the following order: MERC channel (white), transparent mitochondria (pink), opaque mitochondria (red), and transparent ER (green).

**Video 8. 360° views of an individual 95th quantile MERC in a COS-7 cell overexpressing the OMM–ER linker (**Fig. 9**), transfected with ERmoxGFP, labeled for anti-TOM-20, and imaged using 3D STED.** Planar MERCs follow the mitochondria in a single direction with limited travel to additional Z slices. Video was rendered in Imaris 10.0 software using the contacts, mitochondria, and ER channels outputted from MCS-DETECT (50 frames/s). All channels depict a single contact. Additional channels are rendered after each rotation in the following order: MERC channel (white), transparent mitochondria (pink), opaque mitochondria (red), and transparent ER (green).

**Video 9. 360° views of an individual 95th quantile MERC in a COS-7 cell overexpressing WT Gp78-FLAG (**Fig. 9**), transfected with ERmoxGFP, labeled for anti-TOM-20 and anti-FLAG, and imaged using 3D STED.** MERCs display a similar phenotype to MERCs observed in HT-1080 cells. Video was rendered in Imaris 10.0 software using the contacts, mitochondria, and ER channels outputted from MCS-DETECT (50 frames/s). All channels depict a single contact. Additional channels are rendered after each rotation in the following order: MERC channel (white), transparent mitochondria (pink), opaque mitochondria (red), and transparent ER (green).

Video 10. **360° views of an individual 95th quantile MERC in a COS-7 cell overexpressing WT Gp78-FLAG (**Fig. 9**), transfected with ERmoxGFP, labeled for anti-TOM-20 and anti-FLAG, and imaged using 3D STED.** MERCs display a similar phenotype to MERCs observed in HT-1080 cells. Video was rendered in Imaris 10.0 software using the contacts, mitochondria, and ER channels outputted from MCS-DETECT (50 frames/s). All channels depict a single contact. Additional channels are rendered after each rotation in the following order: MERC channel (white), transparent mitochondria (pink), opaque mitochondria (red), and transparent ER (green).

Video 11. **360° views of an individual 95th quantile MERC in a HeLa cells overexpressing WT Gp78-FLAG (**Fig. 9**), transfected with ERmoxGFP, labeled for anti-TOM-20 and anti-FLAG, and imaged using 3D STED.** MERCs display a similar phenotype to MERCs observed in HT-1080 cells. Video was rendered in Imaris 10.0 software using the contacts, mitochondria, and ER channels outputted from MCS-DETECT (10 frames/s). All channels depict a single contact. Additional channels are rendered after each rotation in the following order: MERC channel (white), transparent mitochondria (pink), opaque mitochondria (magenta), and transparent ER (green).

Video 12. **360° views of an individual 95th quantile MERC in a HeLa cell overexpressing WT Gp78-FLAG (**Fig. 9**), transfected with ERmoxGFP, labeled for anti-TOM-20 and anti-FLAG, and imaged using 3D STED.** MERCs display a similar phenotype to MERCs observed in HT-1080 cells. Video was rendered in Imaris 10.0 software using the contacts, mitochondria, and ER channels outputted from MCS-DETECT (10 frames/s). All channels depict a single contact. Additional channels are rendered after each rotation in the following order: MERC channel (white), transparent mitochondria (pink), opaque mitochondria (magenta), and transparent ER (green).

