## [Peer Review File · The Journal of Cell Biology]

Membrane contact site detection (MCS-DETECT) reveals dual control of rough mitochondria-ER contacts

Ben Cardoen, Kurt Vandevoorde, Guang Gao, Milene Ortiz-Silva, Parsa Alan, William Liu, Ellie Tiliakou, Adalbert Vogl, Ghassan Hamarneh, and Ivan Nabi

Corresponding Author(s): Ivan Nabi, University of British Columbia and Ghassan Hamarneh, Simon Fraser University

Review Timeline:

Submission Date:	2022-06-23
Editorial Decision:	2022-07-01
Revision Received:	2022-09-20
Editorial Decision:	2022-11-29
Revision Received:	2023-09-05
Editorial Decision:	2023-10-05
Revision Received:	2023-10-14

Monitoring Editor: Laura Lackner

Scientific Editor: Dan Simon

Transaction Report:

DOI: <https://doi.org/10.1083/jcb.202206109>

July 1, 2022

Re: JCB manuscript #202206109

Dr. Ivan R Nabi
University of British Columbia
Cellular and Physiological Sciences
Life Sciences Institute, University of British Columbia
2350 Health Sciences Mall
Vancouver, BC V6T 1Z3
Canada

Dear Dr. Nabi,

Thank you for submitting your Report manuscript entitled "Automatic sub-precision membrane contact site detection identifies convoluted tubular riboMERCs" to Journal of Cell Biology. As part of our normal reviewing procedure, your paper has been evaluated by at least two editors and an editorial statement is provided below. You will see that, in the consensus opinion of our editors, the manuscript is not a good fit for Journal of Cell Biology. We have thus decided not to subject your manuscript to a lengthy external review process. Because Journal of Cell Biology addresses a wide and diverse audience of cell biologists, we must give priority to manuscripts that provide a substantial advance of broad appeal to the cell biology community, even though many others also present interesting and important advances for researchers in a particular field.

Although we regret that we are not able to consider your manuscript for publication in Journal of Cell Biology, Life Science Alliance (<http://www.life-science-alliance.org/>) will send your paper out for review. LSA is our academic editor-led, open access journal launched as a collaboration between RUP, EMBO Press and Cold Spring Harbor Press. If you wish, you can use the link below to initiate an immediate transfer to LSA. We also think your manuscript may be appropriate for Molecular Biology of the Cell or Journal of Cell Science. Although we have not discussed your paper with editors at these journals, you will find the option to easily transfer your manuscript files to either journal at the link.

Link Not Available

I am sorry that our answer on this occasion is not more positive, and I hope that this outcome will not dissuade you from submitting other manuscripts to us in the future.

Thank you for your interest in Journal of Cell Biology.

With kind regards,

Jodi Nunnari
Editor-in-Chief
Journal of Cell Biology

Editorial Statement:

In this study, the authors describe a novel segmentation-free approach "MCS-DETECT" to identify membrane contact sites with STED super resolution microscopy. They were able to identify and characterize MERCs in whole cell 3D volumes of HT-1080 and COS-7 cells, as well as provide further insight into the role of the Gp78 ubiquitin ligase in the regulation of riboMERCs. In considering the study as a JCB Report, the criteria for this format are that the study represents a highly novel observation that will open up new research avenues. Given what is already known regarding the morphology of MERCs in HT-1080 and COS-7 cells, the observations do not appear to have the novelty or broad interest required for a JCB Report. While it is interesting that the Gp78 ubiquitin ligase activity is important for the tubular morphology of riboMERCs, this observation would need to be investigated in greater mechanistic detail to meet the requirements for novel biological insight, if the study were to be considered for the JCB Tools format. It would be beneficial if the authors provided a more thorough discussion of the submitted paper from Alan et al, for example in the cover letter, to better outline the pertinent findings in that study and the advance represented by the current study in comparison. Altogether, unfortunately the paper seems to not represent the level of advance generally expected to fare well in the JCB peer review process and is therefore being returned to the authors.

November 29, 2022

Re: JCB manuscript #202206109R-A

Dr. Ivan R Nabi
University of British Columbia
Cellular and Physiological Sciences
Life Sciences Institute, University of British Columbia
2350 Health Sciences Mall
Vancouver, BC V6T 1Z3
Canada

Dear Dr. Nabi,

Thank you for submitting your manuscript entitled "Membrane contact site detection (MCS-DETECT) reveals dual control of rough mitochondria-ER contacts." Your manuscript has been assessed by expert reviewers, whose comments are appended below. Although the reviewers express potential interest in this work, significant concerns unfortunately preclude publication of the current version of the manuscript in JCB.

The reviewers all find that MCS-DETECT is an innovative method that provides a significant improvement over current EM-based approaches for analysis of ER-mitochondria contact sites. They do however feel that additional insight into how the different MERC subtypes operate and are structured would enhance the impact of the work. JCB Tools papers are expected to provide a significant technological improvement over existing methods and novel cell biological insight as a proof of principle. Therefore, an in-depth study of the functional differences between the MERC subtypes and a mechanistic explanation of how Gp78 and RRP1 function would not be required. However, we do agree with the reviewers that additional data and controls to clarify the effects of depletions and overexpressions of Gp78 and RRP1 on MERCs are essential. It would also be important to validate MCS-DETECT in other cell lines and quantify the effects of treatments that are known to alter MERCs, such as those suggested by Reviewer #1. Reviewer #3 notes that proximity of the ER to mitochondria does not necessarily imply functional domains, this is an important concern that should be fully addressed. Reviewer #1 asks for more details regarding the mitochondrial vesicles, which are a very intriguing observation but seem to be beyond the scope of this paper, so this will not be required for revision.

Please let us know if you are able to address the major issues outlined above and wish to submit a revised manuscript to JCB. Note that a substantial amount of additional experimental data likely would be needed to satisfactorily address the concerns of the reviewers. The typical timeframe for revisions is three to four months. While most universities and institutes have reopened labs and allowed researchers to begin working at nearly pre-pandemic levels, we at JCB realize that the lingering effects of the COVID-19 pandemic may still be impacting some aspects of your work, including the acquisition of equipment and reagents. Therefore, if you anticipate any difficulties in meeting this aforementioned revision time limit, please contact us and we can work with you to find an appropriate time frame for resubmission. Please note that papers are generally considered through only one revision cycle, so any revised manuscript will likely be either accepted or rejected.

If you choose to revise and resubmit your manuscript, please also attend to the following editorial points. Please direct any editorial questions to the journal office.

GENERAL GUIDELINES:

Text limits: Character count is < 40,000, not including spaces. Count includes title page, abstract, introduction, results, discussion, and acknowledgments. Count does not include materials and methods, figure legends, references, tables, or supplemental legends.

Figures: Your manuscript may have up to 10 main text figures. To avoid delays in production, figures must be prepared according to the policies outlined in our Instructions to Authors, under Data Presentation, <https://jcb.rupress.org/site/misc/ifora.xhtml>. All figures in accepted manuscripts will be screened prior to publication.

Supplemental information: There are strict limits on the allowable amount of supplemental data. Your manuscript may have up to 5 supplemental figures. Up to 10 supplemental videos or flash animations are allowed. A summary of all supplemental material should appear at the end of the Materials and methods section.

Please note that JCB now requires authors to submit Source Data used to generate figures containing gels and Western blots with all revised manuscripts. This Source Data consists of fully uncropped and unprocessed images for each gel/blot displayed in the main and supplemental figures. Since your paper includes cropped gel and/or blot images, please be sure to provide one Source Data file for each figure that contains gels and/or blots along with your revised manuscript files. File names for Source Data figures should be alphanumeric without any spaces or special characters (i.e., SourceDataF#, where F# refers to the associated main figure number or SourceDataFS# for those associated with Supplementary figures). The lanes of the gels/blots should be labeled as they are in the associated figure, the place where cropping was applied should be marked (with a box), and molecular weight/size standards should be labeled wherever possible. Source Data files will be made available to reviewers during evaluation of revised manuscripts and, if your paper is eventually published in JCB, the files will be directly linked to specific figures in the published article.

If you choose to resubmit, please include a cover letter addressing the reviewers' comments point by point. Please also highlight all changes in the text of the manuscript.

Regardless of how you choose to proceed, we hope that the comments below will prove constructive as your work progresses. We would be happy to discuss them further once you've had a chance to consider the points raised. You can contact the journal office with any questions, cellbio@rockefeller.edu or call (212) 327-8588.

Thank you for thinking of JCB as an appropriate place to publish your work.

Sincerely,

Laura Lackner, PhD
Monitoring Editor
Journal of Cell Biology

Dan Simon, PhD
Scientific Editor
Journal of Cell Biology

Reviewer #1 (Comments to the Authors (Required)):

This manuscript presents data from a detailed study of mitochondria-ER contact sites (MERCs). The authors first confirm the existence of a variety of cell-type specific formations of MERCs, either ribosome-studded or smooth. Smooth contacts tend to be closer but for both types, variable widths exist. Maybe more strikingly, using HT-1080 and COS-7 cells, the authors show that drastically different lengths of these membrane contact sites exist. Likewise, the percentage of rough MERCs varies between the two cell lines as well.

The strength of the paper lies in the development of light microscopic tools to assay for the kind and number/extent of different MERCs. To this detail, this could previously only be done through electron-microscopic studies. To achieve this goal, the authors use Gp78 and RRB1 interference, thus giving the study some level of biological significance. They also show nicely the association of RRB1 in the formation of RiboMERCs. Data is presented on the role of Gp78 for tubular MERCs in HT-1080 cells. In this case, overall number of MERCs decreased, but now RRB1 played less of a role. The strength of the study lies in the groundbreaking development of light microscopy-based monitoring of MERC subtypes and their precise quantification. As such, this approach is of high general interest. However at the moment, the study is compromised by a lack of direction of biological experiments and no connections to known MERC modulatory conditions. Also, the choice of cell lines appears problematic in some cases. For instance, is Gp78 expression really tied to the formation of tubular MERCs? Are there physiological conditions that trigger the formation of these MERCs?

Specific points

1. The authors show multinucleated cells in Figure 2 for HT-1080. Is this a general feature? This compromises the findings somewhat, as such cells are typically considered unhealthy. This could be a problem, since cell stress is a known modulator of MERCs, see for instance Csordas et al. 2006. To address this issue, studies should be added using a less abnormal cell line, maybe MCF10A.
2. The detection of putative mitochondrial vesicles is intriguing. At the moment, to me their detection appear to be a missed opportunity, where the authors could have investigated them separately to gain information about their apposition to the ER.
3. The knockout and overexpression studies on Gp78 are done in a curiously selective way, where the respective interference with this protein is only done in one cell type. There is therefore information missing regarding the relative plasticity of MERCs in the two cell lines, dependent on Gp78 (also versus other cell lines, see above).

4. The role of Gp78 remains especially confusing in Figure 5, where it is hard to follow the reasoning of the authors. Here, Gp78 KO decreases MERCs but leaves their relative classification intact (5C). The same is true for RRBp1, which however is mentioned as critical for riboMERC formation.
5. Using Ubc, an OMM-ER linker in addition to Gp78 manipulation, the authors demonstrate distinct shapes for MERCs. This is a proof of principle experiment, not without merit, that is however not used to maximum effect. It would be more informative to test a couple of known conditions that manipulate MERCs through lipids, calcium or oxidative stress to see if one or another MERC type (compact, wide or tubular) proliferates under specific conditions.
6. Can a functional role for subtypes of MERCs be identified? As presented, the tubular MERCs, to give just one example, remain an interesting observation without a clear function, especially versus other types of MERCs.

Minor points:

1. Figure 6B lacks the labels for the conditions used, making interpretation and understanding difficult.

Reviewer #2 (Comments to the Authors (Required)):

In "Membrane contact site detection (MCS-DETECT) reveals dual control of rough mitochondria-ER contacts" the authors describe the development and validation of "MCS-DETECT", an algorithm designed to measure membrane contact sites in 3D super resolution fluorescent images. Specifically in this manuscript, MCS-DETECT is used to measure different types of mitochondria-ER contacts (MERCs) in 3D STED images. Results from MCS-DETECT are compared to results from traditional EM imaging as well as previously published results as validation. Utilizing MCS-DETECT, the authors demonstrate differential effects of Gp78 or RRBp1 loss on the size, shape, and number of MERCs, specifically riboMERCs (ribosome-studded mitochondria-ER contacts) which may indicate a dual regulatory mechanism of riboMERCs.

The MCS-DETECT tool described has great potential for membrane contact analysis. Currently, measuring membrane contacts is either very low-throughput and laborious ("gold standard" EM imaging) or, when it comes to fluorescent imaging, hindered by issues of poor accuracy/reproducibility due to the resolution limit of light, subjectivity/variability in membrane segmentation approaches, and undesirable side-effects of expressing bifluorescent complementation systems. Being able to bypass the need for EM imaging, segmentation, or expression of bifluorescent linkers is a huge boon. The differential effects of altering Gp78 and RRBp1 levels may also have important biological implications barring some issues listed below.

- Data describing the results of RRBp1 knock-down are included in Figures 5-7. These results show that RRBp1 KD causes a dramatic reduction in MERCs and is part of the basis for the authors' conclusion that RRBp1 and Gp78 work in tandem to regulate MERCs. However, there are no controls included demonstrating either efficient knock-down of RRBp1 or controlling for possible off-target effects of the siRNA (i.e. rescue or multiple siRNAs).
- Based on the results in Figure 7A showing a decrease in size but no change in number of large contacts between control and Gp78 KO HT-1080 cells, the authors conclude that Gp78 is responsible for modulating the size of riboMERCs rather than the overall number of riboMERCs. However, both the EM data (Figure 5B) and MCS-DETECT data (Figure 6B) shown earlier in the manuscript display fewer MERCs in Gp78 KO compared to control HT-1080 cells. Is there an explanation for this discrepancy?
- The potential finding that RRBp1 modulates riboMERC formation while Gp78 regulates riboMERC size is interesting, but the biological significance of this finding is not expounded upon.
- There appear to be some formatting issues in Figure 6. In 6B there are no x-axis labels indicating which condition is which. The statistical significance is also very small and partially overlapping the data. 6C which should be displaying KDE plots is referenced in the legend for Figure 6 but these plots are not present in the figure.

Minor issues:

- No n-values are provided for the quantification in Figures 1 and 5.
- In the "Detecting riboMERCs in HT-1080 cells with MCS-DETECT" section pairwise analyses are described but there is not a reference to the figure where these data are presented (presumably Figure 3C).
- In Figures 3A and 4A the boxes indicating the ROIs seem to be slightly misplaced relative to the ROIs shown. Figure 6A is also missing boxes indicating the ROIs.
- In the "Tubular, Gp78-dependent riboMERCs" section "highly significant differences" are described between the largest MERCs in various conditions. These differences are plotted in Figure 7A but there are no statistics included.
- Immunofluorescence images and volume renders are shown in red and green which is not colorblind-friendly.

Overall, this manuscript describes a very useful tool with potential for broad applications (used here with MERCs and STED but with the possibility to be used to analyze other membrane contacts in fluorescent super resolution imaging data). There is sufficient detail provided in the methods section as well as the linked Github repository for others to use this algorithm for their own studies.

Reviewer #3 (Comments to the Authors (Required)):

In this manuscript, Cardoen et al report a careful characterization of mitochondria-ER contacts or MERC in two cell types, HT-1080 and Cos-7. For that, these authors have applied a novel differential analysis, called MCS-DETECT, able to reconstruct ER-

mitochondria contacts at subpixel resolution.

This work displays a very elegant analysis of imaging data to characterize the structure organelle these contacts. Unfortunately, while the method of analysis is quite innovative, these data lack the novelty and rigor required for a JCB publication.

First and foremost, ER-mitochondria contacts are functional domains. The field has demonstrated on several occasions that a mere apposition between ER and mitochondria does not provide with meaningful data on the functional crosstalk between organelles. Therefore, imaging data is insufficient to assess relevance. This can also be applied to any conclusion derived from the distance between these organelles.

Second, recent evidence has shown that ER-mitochondria contacts are altered in cancer, so perhaps data from two cell lines with a conspicuous tubular ER shape should not be extrapolated to the overall regulation of ER-mitochondria contacts or the existence of one or more type of contacts.

Finally, the use of linkers is not quite novel.

Overall, although this report provides with an excellent tool for the analysis of imaging data, it does not provide with meaningful functional readouts. Moreover, relying exclusively on imaging data to draw mechanistic conclusion could end up leading the field astray in the study of these functional contacts.

JCB manuscript #202206109

"Membrane contact site detection (MCS-DETECT) reveals dual control of rough mitochondria-ER contacts".

Cardoen et al

Response to Reviewer Comments

Reviewer #1 (Comments to the Authors (Required)):

This manuscript presents data from a detailed study of mitochondria-ER contact sites (MERCs). The authors first confirm the existence of a variety of cell-type specific formations of MERCs, either ribosome-studded or smooth. Smooth contacts tend to be closer but for both types, variable widths exist. Maybe more strikingly, using HT-1080 and COS-7 cells, the authors show that drastically different lengths of these membrane contact sites exist. Likewise, the percentage of rough MERCs varies between the two cell lines as well.

The strength of the paper lies in the development of light microscopic tools to assay for the kind and number/extent of different MERCs. To this detail, this could previously only be done through electron-microscopic studies. To achieve this goal, the authors use Gp78 and RRB1 interference, thus giving the study some level of biological significance. They also show nicely the association of RRB1 in the formation of RiboMERCs. Data is presented on the role of Gp78 for tubular MERCs in HT-1080 cells. In this case, overall number of MERCs decreased, but now RRB1 played less of a role. The strength of the study lies in the groundbreaking development of light microscopy-based monitoring of MERC subtypes and their precise quantification. As such, this approach is of high general interest. However at the moment, the study is compromised by a lack of direction of biological experiments and no connections to known MERC modulatory conditions. Also, the choice of cell lines appears problematic in some cases. For instance, is Gp78 expression really tied to the formation of tubular MERCs? Are there physiological conditions that trigger the formation of these MERCs?

Specific points

1. The authors show multinucleated cells in Figure 2 for HT-1080. Is this a general feature? This compromises the findings somewhat, as such cells are typically considered unhealthy. This could be a problem, since cell stress is a known modulator of MERCs, see for instance Csordas et al. 2006. To address this issue, studies should be added using a less abnormal cell line, maybe MCF10A.

Multinucleation is not a typical feature of HT-1080 cells and we have replaced the image stacks for both HT-1080 and COS-7 cells in Fig. 2A. The cells used in this study, HT-1080, COS-7 and now HeLa, are all cancer cells and a minority of the cells may be multinucleated. HT-1080 cells are the only cell line that we are aware of that natively express riboMERCs, with other studies of native expression of riboMERCs performed in liver (Anastasia et al., 2021; Csordas et al., 2006). We understand that MERCs are altered in cancer cells and in response to stress and are using HT-1080 cells as a cell model to develop a tool to detect MERCs. Future studies of riboMERC expression and function are

certainly envisaged, including their differential distribution between normal and cancer cells, and will benefit from the ability to detect these MERCs by fluorescent microscopy.

2. The detection of putative mitochondrial vesicles is intriguing. At the moment, to me their detection appear to be a missed opportunity, where the authors could have investigated them separately to gain information about their apposition to the ER.

We agree that interaction of putative mitochondrial vesicles with the ER is of potential interest. We excluded the vesicles from the analysis as the corresponding EM quantification was based on interaction of ER with whole mitochondria. We believe that further experimentation, beyond TOM20 labeling is required to define mitochondrial vesicles and their interaction with ER. We have however now modified Supp. Fig. 2 to include the ER channel to enable the reader to assess the extent of interaction with the ER of the mitochondrial vesicles that we are filtering out.

3. The knockout and overexpression studies on Gp78 are done in a curiously selective way, where the respective interference with this protein is only done in one cell type. There is therefore information missing regarding the relative plasticity of MERCs in the two cell lines, dependent on Gp78 (also versus other cell lines, see above).

HT-1080 cells were chosen for the knockout studies as we had previously shown that they express Gp78-dependent riboMERCs (Wang et al., 2015). We chose COS-7 for the overexpression studies because our EM data (Figure 1) showed they have no elongated riboMERCs, in addition to the fact that this cell line is amenable to transfection and imaging. The ability of Gp78 overexpression to induce morphologically similar riboMERCs in COS-7 to those lost upon Gp78 knockout and RRBP1 knockdown in HT-1080 cells highlights the common nature of riboMERCs in these two cell lines. We have now added analysis of Gp78, the Gp78 Ring finger mutant, and the OMM-ER linker in HeLa cells (Figs. 5, 8, 9). These results show that Gp78 expression in HeLa cells induces large, tubular riboMERCs morphologically similar to those present in HT-1080 cells and induced by Gp78 expression in COS-7 cells. Interestingly, the riboMERCs induced by Gp78 in HeLa cells were smaller than those induced in COS-7 cells and the OMM-ER linker did not induce the same increase in MERCs that we observed in COS-7. The HeLa and COS-7 data sets were acquired separately, and to test whether expression levels of the constructs differed between the two cell lines we transfected the constructs into the two cell lines in parallel. As can be seen in Supp. Fig. 3, both Flag-Gp78 and the OMM-ER-linker are significantly more highly expressed in COS-7 than in HeLa cells, explaining the differential results obtained with these two constructs. Curiously, we did not observe increased expression in COS-7 cells for the Gp78 RM mutant. Overexpression of the Gp78 RM mutant shows higher expression levels than WT Gp78 and induces dramatic changes in ER morphology (St-Pierre et al., 2012); we consider that it is likely that very high expression of the Gp78 RM mutant is cytotoxic. We also observed larger MERCs upon expression of the Gp78 RM mutant in HeLa cells however shape features of these structures showed that they are distinct from the tubular MERCs induced by Gp78 (Supp. Fig. 3).

4. The role of Gp78 remains especially confusing in Figure 5, where it is hard to follow the reasoning of the authors. Here, Gp78 KO decreases MERCs but leaves their relative classification intact (5C). The same is true for RRBP1, which however is mentioned as critical for riboMERC formation.

Figure 5 (now 6) shows that Gp78 KO reduces the abundance of riboMERCs by half and RRBP1 knockdown by about 90% in HT-1080 cells, but does not eliminate them. Consistently, those structures classified as riboMERCs based on the presence of ribosomes in the interorganellar space, even if few in number, retain the same increased contact width relative to smooth ER mitochondria contacts. These results are consistent with our prior publication showing that Gp78 knockdown only partially reduces the expression of riboMERCs (Wang et al., 2015). And while our results support an essential role for RRBP1 in riboMERC formation, knockdown efficiency is perhaps 90% such that not all riboMERCs will necessarily be eliminated (Supp. Fig. 3B). We consider that the constant width observed for riboMERCs, defined morphologically by the presence of ribosomes in the inter-organellar space, validates our classification of these distinct MERCs by EM. We conclude that while Gp78 KO and siRRBP1 reduce the abundance of riboMERCs, their morphological classification (presence of interorganellar ribosomes and increased width between ER and mitochondrial membranes) remains intact.

5. Using Ubc, an OMM-ER linker in addition to Gp78 manipulation, the authors demonstrate distinct shapes for MERCs. This is a proof of principle experiment, not without merit, that is however not used to maximum effect. It would be more informative to test a couple of known conditions that manipulate MERCs through lipids, calcium or oxidative stress to see if one or another MERC type (compact, wide or tubular) proliferates under specific conditions.

These are excellent ideas and we have now extended our previous study that Gp78 regulates mitochondrial potential to show that: 1) reduction of riboMERCs, through either Gp78 KO or RRBP1 knockdown reduces mitochondrial potential; and 2) that induction of riboMERCs through overexpression of Gp78, but not the Gp78 Ring finger mutant or the OMM-ER linker, in COS-7 and HeLa cells increases mitochondrial potential. We are not aware of any treatment conditions known to specifically impact riboMERCs such that any such studies would be exploratory in nature and beyond the scope of the current study. Use of MCS-DETECT to characterize the impact of calcium, lipids, oxidative stress and other treatment conditions on MERC expression is certainly of interest and will be explored in future studies. For instance, preliminary results extending our prior 3D studies of SARS-CoV-2 infected cells (Shapira et al., 2022) now shows that use of MCS-DETECT detects increased MERC expression in SARS-CoV-2 infected cells (study in progress).

6. Can a functional role for subtypes of MERCs be identified? As presented, the tubular MERCs, to give just one example, remain an interesting observation without a clear function, especially versus other types of MERCs.

Morphological classification of MERC subtypes beyond smooth and riboMERCs remains a challenge. Indeed, the distinct morphology of riboMERCs was the basis for our choice of

this class of MERCs to validate MCS-DETECT MERC detection. As mentioned above, we believe that the ability to detect this distinct class of MERCs with MCS-DETECT and modify their expression using specific regulators and tethers will now allow us to more specifically explore the function of this MERC subtype. We have now included data showing that reduced expression of riboMERCs, via modulation of expression of either Gp78 or RRB1, impacts mitochondrial potential. This is consistent with our recent publication showing that Gp78 promotes mitophagy and mitochondrial health (Alan et al., 2022). We are actively exploring the role of riboMERCs in mitophagy as well as in lipid metabolism, based on the work of the Pellegrini lab in liver (Anastasia et al., 2021; Ilacqua et al., 2022). We consider that these more functional experiments are beyond the scope of this JCB Tools article.

Minor points:

1. Figure 6B lacks the labels for the conditions used, making interpretation and understanding difficult.

Labels for Figure 6B (now figure 7) have been added.

Reviewer #2 (Comments to the Authors (Required)):

In "Membrane contact site detection (MCS-DETECT) reveals dual control of rough mitochondria-ER contacts" the authors describe the development and validation of "MCS-DETECT", an algorithm designed to measure membrane contact sites in 3D super resolution fluorescent images. Specifically in this manuscript, MCS-DETECT is used to measure different types of mitochondria-ER contacts (MERCs) in 3D STED images. Results from MCS-DETECT are compared to results from traditional EM imaging as well as previously published results as validation. Utilizing MCS-DETECT, the authors demonstrate differential effects of Gp78 or RRB1 loss on the size, shape, and number of MERCs, specifically riboMERCs (ribosome-studded mitochondria-ER contacts) which may indicate a dual regulatory mechanism of riboMERCs.

The MCS-DETECT tool described has great potential for membrane contact analysis. Currently, measuring membrane contacts is either very low-throughput and laborious ("gold standard" EM imaging) or, when it comes to fluorescent imaging, hindered by issues of poor accuracy/reproducibility due to the resolution limit of light, subjectivity/variability in membrane segmentation approaches, and undesirable side-effects of expressing bifluorescent complementation systems. Being able to bypass the need for EM imaging, segmentation, or expression of bifluorescent linkers is a huge boon. The differential effects of altering Gp78 and RRB1 levels may also have important biological implications barring some issues listed below.

- Data describing the results of RRB1 knock-down are included in Figures 5-7. These results show that RRB1 KD causes a dramatic reduction in MERCs and is part of the basis for the authors' conclusion that RRB1 and Gp78 work in tandem to regulate MERCs. However, there are no controls included demonstrating either efficient knock-down of

RRBP1 or controlling for possible off-target effects of the siRNA (i.e. rescue or multiple siRNAs).

We have now included a Western blot showing efficiency of RRBP1 knockdown (Supp. Fig. 3B).

- Based on the results in Figure 7A showing a decrease in size but no change in number of large contacts between control and Gp78 KO HT-1080 cells, the authors conclude that Gp78 is responsible for modulating the size of riboMERCs rather than the overall number of riboMERCs. However, both the EM data (Figure 5B) and MCS-DETECT data (Figure 6B) shown earlier in the manuscript display fewer MERCs in Gp78 KO compared to control HT-1080 cells. Is there an explanation for this discrepancy? Gp78 KO reduces both the size and number of large contacts in HT-1080 cells.

In response to this comment, we highlight that the global (i.e. whole cell analysis) EM and MCS-DETECT data are consistent with each other. The fact that the number of large contacts (> than 500 pixels, the average size of the largest 5% of MERCs) is similar but that the size of the top 5% largest (Q95) of contacts is reduced suggests change occurs in the largest contacts (5%), something that is non-trivial to capture in a stable way by looking at mean or fixed sizes. For instance, breaking of a large contact into two smaller contacts, still larger than >500 pixels, will increase the number of large contacts and reduce the overall size of the largest contacts. This is consistent with the EM data showing that the length of riboMERCs by EM and size by MCS-DETECT is reduced upon Gp78 KO.

- The potential finding that RRBP1 modulates riboMERC formation while Gp78 regulates riboMERC size is interesting, but the biological significance of this finding is not expounded upon.

We have now included data showing that regulation of riboMERCs impacts mitochondrial potential (Figs. 4, 5, 7). This demonstrates physiological relevance for riboMERCs and further studies will be undertaken to define the mechanism, focusing initially on riboMERC regulation of mitophagy (Alan et al., 2022).

- There appear to be some formatting issues in Figure 6. In 6B there are no x-axis labels indicating which condition is which. The statistical significance is also very small and partially overlapping the data. 6C which should be displaying KDE plots is referenced in the legend for Figure 6 but these plots are not present in the figure.

Figure 6B (now 7B) has been corrected and legend updated to no longer refer to KDE plots.

Minor issues:

- No n-values are provided for the quantification in Figures 1 and 5.

N values for the EM studies have been added to Figure legends 1 and 5 (now Fig. 6).

- In the "Detecting riboMERCs in HT-1080 cells with MCS-DETECT" section pairwise analyses are described but there is not a reference to the figure where these data are presented (presumably Figure 3C).

The reference to Fig 3C has been added to the text.

- In Figures 3A and 4A the boxes indicating the ROIs seem to be slightly misplaced relative to the ROIs shown. Figure 6A is also missing boxes indicating the ROIs.

The placement of the ROIs has been corrected.

- In the "Tubular, Gp78-dependent riboMERCs" section "highly significant differences" are described between the largest MERCs in various conditions. These differences are plotted in Figure 7A but there are no statistics included.

The p value indicators for Figure 7 (now Figure 8) have been updated.

- Immunofluorescence images and volume renders are shown in red and green which is not colorblind-friendly.

All images have now been rendered using magenta for the mitochondria channel.

Overall, this manuscript describes a very useful tool with potential for broad applications (used here with MERCs and STED but with the possibility to be used to analyze other membrane contacts in fluorescent super resolution imaging data). There is sufficient detail provided in the methods section as well as the linked Github repository for others to use this algorithm for their own studies.

Reviewer #3 (Comments to the Authors (Required)):

In this manuscript, Cardoen et al report a careful characterization of mitochondria-ER contacts or MERC in two cell types, HT-1080 and Cos-7. For that, these authors have applied a novel differential analysis, called MCS-DETECT, able to reconstruct ER-mitochondria contacts at subpixel resolution.

This work displays a very elegant analysis of imaging data to characterize the structure organelle these contacts. Unfortunately, while the method of analysis is quite innovative, these data lack the novelty and rigor required for a JCB publication.

First and foremost, ER-mitochondria contacts are functional domains. The field has demonstrated on several occasions that a mere apposition between ER and mitochondria does not provide with meaningful data on the functional crosstalk between organelles. Therefore, imaging data is insufficient to assess relevance. This can also be applied to any conclusion derived from the distance between these organelles.

Yes, mere apposition of two organelles is not necessarily sufficient for functional crosstalk between organelles. However, MERCs are not just functional domains but also

morphological domains, and it is our expectation that MCS-DETECT may provide a means to attribute specific MERC functions to specific MERC subtypes. We now include data showing that Gp78 and RRBP1 regulation of riboMERCs is associated with reduced mitochondrial potential supporting functional relevance of these MERCs. We agree that specific roles of these structures remain to be determined not to mention the functional relevance of the distance between the organelles. Future application of this approach to determine 3D structure of MERCs in live 3D image stacks, using cellular models and fluorescent reporters that report on functional roles of MERCs and other organelle contact sites may lead to findings of functional relevance and enhance our understanding of the relationship between MERC morphology, diversity and function.

Second, recent evidence has shown that ER-mitochondria contacts are altered in cancer, so perhaps data from two cell lines with a conspicuous tubular ER shape should not be extrapolated to the overall regulation of ER-mitochondria contacts or the existence of one or more type of contacts.

We do not suggest to extrapolate our data on riboMERCs to overall regulation of MERCs. Rather, we are using a specific cell model (HT-1080), shown to express at least two types of MERCs by EM, to study the molecular regulation (Gp78, RRBP1) of a distinct contact site as a test case to evaluate the ability of MCS-DETECT to detect MERCs. Existence of smooth and riboMERCs in normal cells is well-established based on earlier work in liver from the Hajnockzy and Pellegrini labs (Anastasia et al., 2021; Csordas et al., 2006). While the tubular MERCs we report here are not morphologically identical to the wrappER reported in liver by Pellegrini, they may represent a less-developed form of the same contact.

Finally, the use of linkers is not quite novel.

Our use of linkers was not intended to be novel but to provide a means to show that MCS-DETECT is able to detect increased MERCs due to linker expression.

Overall, although this report provides with an excellent tool for the analysis of imaging data, it does not provide with meaningful functional readouts. Moreover, relying exclusively on imaging data to draw mechanistic conclusion could end up leading the field astray in the study of these functional contacts.

We thank the reviewer for their acknowledgement of the strength of our tool and agree that on its own it cannot provide meaningful functional readouts. In light of the extensive use in the field of colocalization using diffraction limited microscopy to assess contact formation, we argue that providing a validated sub-resolution fluorescent tool to detect contacts to be used in parallel with EM and the various functional assays available to study MERCs should enable determination of how and whether MERC diversity (morphological and molecular) impact MERC expression, morphology and function.

- Alan, P., K.R. Vandevoorde, B. Joshi, B. Cardoen, G. Gao, Y. Mohammadzadeh, G. Hamarneh, and I.R. Nabi. 2022. Basal Gp78-dependent mitophagy promotes mitochondrial health and limits mitochondrial ROS. *Cell Mol Life Sci.* 79:565.
- Anastasia, I., N. Ilacqua, A. Raimondi, P. Lemieux, R. Ghandehari-Alavijeh, G. Faure, S.L. Mekhedov, K.J. Williams, F. Caicci, G. Valle, M. Giacomello, A.D. Quiroga, R. Lehner, M.J. Miksis, K. Toth, T.Q. de Aguiar Vallim, E.V. Koonin, L. Scorrano, and L. Pellegrini. 2021. Mitochondria-rough-ER contacts in the liver regulate systemic lipid homeostasis. *Cell Rep.* 34:108873.
- Csordas, G., C. Renken, P. Varnai, L. Walter, D. Weaver, K.F. Buttle, T. Balla, C.A. Mannella, and G. Hajnoczky. 2006. Structural and functional features and significance of the physical linkage between ER and mitochondria. *J Cell Biol.* 174:915-921.
- Ilacqua, N., I. Anastasia, D. Alosyn, R. Ghandehari-Alavijeh, E.A. Peluso, M.C. Brearley-Sholto, L.V. Pellegrini, A. Raimondi, T.Q. de Aguiar Vallim, and L. Pellegrini. 2022. Expression of Synj2bp in mouse liver regulates the extent of wrapER-mitochondria contact to maintain hepatic lipid homeostasis. *Biology Direct.* 17:37.
- Shapira, T., I.A. Monreal, S.P. Dion, D.W. Buchholz, B. Imbiakha, A.D. Olmstead, M. Jager, A. Desilets, G. Gao, M. Martins, T. Vandal, C.A.H. Thompson, A. Chin, W.D. Rees, T. Steiner, I.R. Nabi, E. Marsault, J. Sahler, D.G. Diel, G.R. Van de Walle, A. August, G.R. Whittaker, P.L. Boudreault, R. Leduc, H.C. Aguilar, and F. Jean. 2022. A TMPRSS2 inhibitor acts as a pan-SARS-CoV-2 prophylactic and therapeutic. *Nature.* 605:340-348.
- St-Pierre, P., T. Dang, B. Joshi, and I.R. Nabi. 2012. Peripheral endoplasmic reticulum localization of the Gp78 ubiquitin ligase activity. *J Cell Sci.* 125:1727-1737.
- Wang, P.T., P.O. Garcin, M. Fu, M. Masoudi, P. St-Pierre, N. Pante, and I.R. Nabi. 2015. Distinct mechanisms controlling rough and smooth endoplasmic reticulum contacts with mitochondria. *J Cell Sci.* 128:2759-2765.

October 5, 2023

RE: JCB Manuscript #202206109RR

Dr. Ivan R Nabi
University of British Columbia
Cellular and Physiological Sciences
Life Sciences Institute, University of British Columbia
2350 Health Sciences Mall
Vancouver, BC V6T 1Z3
Canada

Dear Dr. Nabi,

Thank you for submitting your revised manuscript entitled "Membrane contact site detection (MCS-DETECT) reveals dual control of rough mitochondria-ER contacts." The manuscript has been re-assessed by all of the original reviewers. We would be happy to publish your paper in JCB pending final revisions necessary to address the remaining reviewer points as we as to meet our formatting guidelines (see details below).

The reviewers ask for a Western blot showing Gp78 expression levels, images to go along with measurements of mitochondrial membrane potential, and several minor text and figure changes. As this is a Tools paper additional experiments to investigate mechanism will not be necessary. However, we ask that you please tone down claims regarding a "dual mechanism" by replacing this with "process" or something similar.

A. MANUSCRIPT ORGANIZATION AND FORMATTING:

1) Text limits: Character count for Tools is < 40,000, not including spaces. Count includes title page, abstract, introduction, results, discussion, and acknowledgments. Count does not include materials and methods, figure legends, references, tables, or supplemental legends.

2) Figure formatting: Tools may have up to 10 main text figures. Scale bars must be present on all microscopy images, including inset magnifications. Please add scale bars to magnifications in Figures 1A/B & S2B(COS7 before image). Molecular weight or nucleic acid size markers must be included on all gel electrophoresis. Please add the size markers to blots in Figure S3B.

Also, please avoid pairing red and green for images and graphs to ensure legibility for color-blind readers. If red and green are paired for images, please ensure that the particular red and green hues used in micrographs are distinctive with any of the colorblind types. If not, please modify colors accordingly or provide separate images of the individual channels.

3) Statistical analysis: Error bars on graphic representations of numerical data must be clearly described in the figure legend. The number of independent data points (n) represented in a graph must be indicated in the legend. Please, indicate whether 'n' refers to technical or biological replicates (i.e. number of analyzed cells, samples or animals, number of independent experiments). If independent experiments with multiple biological replicates have been performed, we recommend using distribution-reproducibility SuperPlots (please see Lord et al., JCB 2020) to better display the distribution of the entire dataset, and report statistics (such as means, error bars, and P values) that address the reproducibility of the findings.

Statistical methods should be explained in full in the materials and methods. For figures presenting pooled data the statistical measure should be defined in the figure legends. Please also be sure to indicate the statistical tests used in each of your experiments (both in the figure legend itself and in a separate methods section) as well as the parameters of the test (for example, if you ran a t-test, please indicate if it was one- or two-sided, etc.). Also, if you used parametric tests, please indicate if the data distribution was tested for normality (and if so, how). If not, you must state something to the effect that "Data distribution was assumed to be normal but this was not formally tested."

4) Materials and methods: Should be comprehensive and not simply reference a previous publication for details on how an experiment was performed. Please provide full descriptions (at least in brief) in the text for readers who may not have access to referenced manuscripts. The text should not refer to methods "...as previously described." Please add a section describing the full immunoblotting procedure including the type of membrane used, all antibodies, and acquisition and quantification methods.

5) For all cell lines, vectors, constructs/cDNAs, etc. - all genetic material: please include database / vendor ID (e.g., Addgene, ATCC, etc.) or if unavailable, please briefly describe their basic genetic features, even if described in other published work or gifted to you by other investigators (and provide references where appropriate). Please be sure to provide the sequences for all of your oligos: primers, si/shRNA, RNAi, gRNAs, etc. in the materials and methods. You must also indicate in the methods the source, species, and catalog numbers/vendor identifiers (where appropriate) for all of your antibodies, including secondary. If antibodies are not commercial, please add a reference citation if possible.

6) Microscope image acquisition: The following information must be provided about the acquisition and processing of images:

- a. Make and model of microscope
- b. Type, magnification, and numerical aperture of the objective lenses
- c. Temperature
- d. Imaging medium
- e. Fluorochromes
- f. Camera make and model
- g. Acquisition software
- h. Any software used for image processing subsequent to data acquisition. Please include details and types of operations involved (e.g., type of deconvolution, 3D reconstructions, surface or volume rendering, gamma adjustments, etc.).

7) References: There is no limit to the number of references cited in a manuscript. References should be cited parenthetically in the text by author and year of publication. Abbreviate the names of journals according to PubMed.

8) Supplemental materials: There are strict limits on the allowable amount of supplemental data. Tools may have up to 5 supplemental figures and 10 videos. You currently exceed the video limit but, in this case, we will be able to give you the extra space. Please also note that tables, like figures, should be provided as individual, editable files. A summary of all supplemental material should appear at the end of the Materials and methods section. Please include one brief sentence per item.

9) Videos: JCB requires MP4 files no larger than 20 MB for publication. For optimal compatibility across operating systems and devices, please select H.264 compression when saving. Videos legends should describe what is being shown, the cell type or tissue being viewed (including relevant cell treatments, concentration and duration, or transfection), the imaging method (e.g., time-lapse epifluorescence microscopy), what each color represents, how often frames were collected, the frames/second display rate, and the number of any figure that has related video stills or images.

10) eTOC summary: A ~40-50 word summary that describes the context and significance of the findings for a general readership should be included on the title page. The statement should be written in the present tense and refer to the work in the third person. It should begin with "First author name(s) et al..." to match our preferred style.

11) Conflict of interest statement: JCB requires inclusion of a statement in the acknowledgements regarding competing financial interests. If no competing financial interests exist, please include the following statement: "The authors declare no competing financial interests." If competing interests are declared, please follow your statement of these competing interests with the following statement: "The authors declare no further competing financial interests."

12) A separate author contribution section is required following the Acknowledgments in all research manuscripts. All authors should be mentioned and designated by their first and middle initials and full surnames. We encourage use of the CRediT nomenclature (<https://casrai.org/credit/>).

13) ORCID IDs: ORCID IDs are unique identifiers allowing researchers to create a record of their various scholarly contributions in a single place. Please note that ORCID IDs are required for all authors. At resubmission of your final files, please be sure to provide your ORCID ID and those of all co-authors.

14) JCB requires authors to submit Source Data used to generate figures containing gels and Western blots with all revised manuscripts. This Source Data consists of fully uncropped and unprocessed images for each gel/blot displayed in the main and supplemental figures. Since your paper includes cropped gel and/or blot images, please be sure to provide one Source Data file for each figure that contains gels and/or blots along with your revised manuscript files. File names for Source Data figures should be alphanumeric without any spaces or special characters (i.e., SourceDataF#, where F# refers to the associated main figure number or SourceDataFS# for those associated with Supplementary figures). The lanes of the gels/blots should be labeled as they are in the associated figure, the place where cropping was applied should be marked (with a box), and molecular weight/size standards should be labeled wherever possible.

Source Data files will be directly linked to specific figures in the published article. Source Data Figures should be provided as individual PDF files (one file per figure). Authors should endeavor to retain a minimum resolution of 300 dpi or pixels per inch. Please review our instructions for export from Photoshop, Illustrator, and PowerPoint here: <https://rupress.org/jcb/pages/submission-guidelines#revised>

15) Journal of Cell Biology now requires a data availability statement for all research article submissions. These statements will

be published in the article directly above the Acknowledgments. The statement should address all data underlying the research presented in the manuscript. Please visit the JCB instructions for authors for guidelines and examples of statements at (<https://rupress.org/jcb/pages/editorial-policies#data-availability-statement>).

B. FINAL FILES:

Thank you for this interesting contribution, we look forward to publishing your paper in Journal of Cell Biology.

Sincerely,

Laura Lackner, PhD
Monitoring Editor
Journal of Cell Biology

Dan Simon, PhD
Scientific Editor
Journal of Cell Biology

Reviewer #1 (Comments to the Authors (Required)):

Cardoen et al. have revised their manuscript and the majority of my points have been addressed. A few minor issues remain but this should not detract from my overall positive impression.

Minor Points:

1. The authors should provide a Western blot that allows the reader to understand the distinct Gp78 expression levels associated with Figure 4A/B.
2. The mitochondrial membrane potential is not necessarily an indication of mitochondrial health (regarding Figure 7C). It simply indicates the strength of the proton gradient, but a high level could also indicate that mitochondria do not make use of it, hence could be considered "unhealthy".

Reviewer #2 (Comments to the Authors (Required)):

In this revision, the authors have made the following important changes:

Data related to expression of Grp78, Grp78 mutant, and the OMM linker are now described for HeLa cells as well as COS7 cells. This improves confidence in the generality of the results as similar phenotypes are observed in both cell types.

There are now also experiments showing alterations in mitochondrial membrane potential in response to alterations in MERCs. This demonstrates at least one physiological consequence of altered MERCs and makes the results of the manuscript less phenomenological.

These changes directly address the concerns of reviewers over interpretation of results from a limited number of cell lines and the functional relevance of the alterations in MERCs measured by the MCS-DETECT method.

There are also some improvements to the visual presentation of the images as well as some additional information included such as blots confirming knock-down of RRBP and visual examples of how mitochondria were filtered for analysis.

All these additions/modifications improve the quality and interpretability of the manuscript.

There remain a few minor aspects that could further improve the quality of the manuscript:

Quantification of the alterations in mitochondrial membrane potential are provided but there are no images included. Images would allow readers to gauge if changes in membrane potential are related to a uniform change across all cells or a more heterogeneous effect. Also, the language should be toned down to not overinterpret the changes to mitochondrial membrane potential in terms of functional or physiological implications.

In Figure 9, MERCs are color-coded by size. This is appreciated and a nice way to display the differences but was not immediately clear from looking at the figure. It would be nice if the color-coded scale bar in the figure were labelled such that readers immediately know what is being shown.

Overall, this is a well-done manuscript describing an important tool with potential for broad use in the growing field of mito-ER contacts. There is still yet much to understand about ER-mito contacts, and this paper definitely does not solve all of them. However, tools such as these are essential for enabling further research into these processes. Thus, while this is more of a techniques/tools paper than a "new cell biology" paper, I think this is a very nice fit for JCB and merits publication here.

Reviewer #3 (Comments to the Authors (Required)):

I would like to thank the authors for their efforts to improve this manuscript. However, while I believe the report can advance our understanding on the morphology of MERCs in these cell types, I think it is more suitable for a more technical or specialized journal.

In would like to underscore that ER-mitochondria structural data is still highly dependent on the observer, particularly when using EM images obtained by negative staining. Indeed, some authors have observed that alterations in MERCs and functional ER-mitochondria crosstalk do not correlate. The reason why being that many organelles in the cell are in close contact when imaging data is analyzed, however they do not necessarily have a functional interaction. Therefore, at the moment, there are no data that consistently and rigorously link the degree of physical apposition and functional crosstalk.

For these reasons and emphasizing the innovative nature of the method presented here, it is difficult to be confident that a structural analysis has currently the resolution necessary to be a reporter of ER-mitochondria functional communication. To this regard, I thank the authors for including measurements of mitochondrial membrane potential. However, the methodology used to measure it is not adequate, since it is known in the field that mitoView633, as well as other similar probes, are not as sensitive as publicized when quantifying mitochondrial membrane potential. Nonetheless, the data presented in Fig. 4D could be result of the well-known impact of GP78 on mitochondria biology, rather than the effect on their interaction with the ER.

In summary, it is my opinion that the authors have developed a strong tool for the analysis of imaging data, but I reiterate that drawing mechanistic conclusion from structural data would just contribute to cloud our understanding of these contact sites.